# ZBTB48 is a priming factor regulating B-cell-specific CIITA expression

Grishma Rane [1], Vivian L S Kuan [1], Suman Wang[2], Michelle Meng Huang Mok [1], Vartika Khanchandani [1], Julia Hansen [1], Ieva Norvaisaite[1], Naasyidah Zulkaflee[1], Wai Khang Yong [1], Arne Jahn [3,4,5,6], Vineeth T Mukundan[1], Yunyu Shi[2], Motomi Osato[1], Fudong Li[2] & Dennis Kappei [1,7,8 ✉]

## Abstract

The class-II transactivator (CIITA) is the master regulator of MHC class-II gene expression and hence the adaptive immune response. Three cell type-specific promoters (pI, pIII, and pIV) are involved in the regulation of CIITA expression, which can be induced by IFN-γ in non-immune cells. While key regulatory elements have been identified within these promoters, our understanding of the transcription factors regulating CIITA expression is incomplete. Here, we demonstrate that the telomere-binding protein and transcriptional activator ZBTB48 directly binds to both critical activating elements within the B-cell-specific promoter CIITA pIII. ZBTB48 knockout impedes the CIITA/MHC-II expression program induced in non-APC cells by IFN-γ, and loss of ZBTB48 in mice silences MHC-II expression in pro-B and immature B cells. Transcriptional regulation of CIITA by ZBTB48 is enabled by ZBTB48-dependent chromatin opening at CIITA pIII upstream of activating H3K4me3 marks. We conclude that ZBTB48 primes CIITA pIII by acting as a molecular on-off-switch for B-cell-specific CIITA expression.

**Keywords** MHC II; CIITA; B Cell; Gene Expression; Epigenetics
**Subject Categories** Chromatin, Transcription & Genomics; Immunology

## Introduction

Major histocompatibility complex class-II (MHC II or HLA class-II) cell surface protein complexes are essential for the initiation, amplification, and regulation of adaptive immune responses and the development of an immunological memory. Precise expression and assembly of MHC-II complexes are thus critical for protection against pathogens, curbing tumor growth as well as moderating autoimmune responses (Klein et al, 1993; Swanberg et al, 2005). While MHC II is constitutively expressed on professional antigen-presenting cells (APCs) such as macrophages, dendritic cells and B cells, it can also be induced by cytokines such as interferon-γ (IFNγ) (Pober et al, 1983) in non-APCs (Reith et al, 2005).

The co-ordinated expression of MHC-II genes is regulated by common *cis*-acting DNA elements, which are bound by three direct DNA-binding proteins, RFX (Masternak et al, 1998; Nagarajan et al, 1999; Durand et al, 1997; Steimle et al, 1995), NF-Y (Maity and Crombrugghe, 1998; Mantovani, 1999) and CREB (Moreno et al, 1999), which serve as a combined interaction platform for the MHC-II master regulator CIITA (MHC2TA) (Masternak et al, 2000). As the critical upstream regulator, CIITA foreshadows the MHC-II expression pattern with constitutive expression in APCs and inducible expression in non-APCs, e.g., upregulation upon IFNγ treatment via JAK-STAT signaling (Steimle et al, 1994; Muhlethaler-Mottet et al, 1998). Mutations in CIITA and other MHC-II enhanceosome genes cause the bare lymphocyte syndrome (BLS) (Steimle et al, 1993, 1995; Nagarajan et al, 1999; Masternak et al, 1998; Durand et al, 1997; Reith et al, 1988). In addition, repression of CIITA is a commonly employed mechanism for MHC-II downregulation in diseases with inflammatory components (Swanberg et al, 2005) and by pathogens and cancer cells to avoid immune recognition (Roche and Furuta, 2015; Reith et al, 2005). For instance, CIITA gene fusions that occur frequently in lymphoid cancers concomitantly result in loss of MHC-II expression (Steidl et al, 2011). Overall, CIITA is essential for both constitutive and inducible expression of MHC II as the master regulator of the adaptive immunity gene expression program.

CIITA transcription itself is controlled by a 14 kb multi-promoter regulatory region, containing three independent promoter elements in humans, pI, pIII, and pIV, which are expressed in a tissue-specific manner (Muhlethaler-Mottet et al, 1997). pI is expressed in dendritic cells and macrophages (Muhlethaler-Mottet et al, 1997; Pai et al, 2002), while pIII is responsible for constitutive CIITA expression in B cells, human-activated T cells and plasmacytoid dendritic cells (Muhlethaler-Mottet et al, 1997; Pai et al, 2002; Wong et al, 2002). Although all three promoters can be induced by IFNγ to varying degrees, pIV is considered as the predominant responder to IFNγ-induction (Piskurich et al, 1999;

[1]Cancer Science Institute of Singapore, National University of Singapore, 117599 Singapore, Singapore. [2]MOE Key Laboratory for Cellular Dynamics, School of Life Sciences, Division of Life Sciences and Medicine, University of Science and Technology of China, Hefei, China. [3]Institute for Clinical Genetics, University Hospital Carl Gustav Carus at the Technische Universität Dresden, Dresden, Germany. [4]National Center for Tumor Diseases (NCT), Dresden, Germany. [5]German Cancer Research Center (DKFZ), Heidelberg, Germany. [6]ERN-GENTURIS, Hereditary Cancer Syndrome Center, Dresden, Germany. [7]Department of Biochemistry, Yong Loo Lin School of Medicine, National University of Singapore, 117596 Singapore, Singapore. [8]NUS Center for Cancer Research, Yong Loo Lin School of Medicine, National University of Singapore, Singapore, Singapore. ✉E-mail: dennis.kappei@nus.edu.sg

Waldburger et al, 2001). B-cell-specific CIITA expression depends on a 319 bp proximal promoter element (Ghosh et al, 1999; Stoep et al, 2002). Prior genomic footprint analysis had identified two sequences, activation response element (ARE) 1 and 2, that are essential for CIITA expression in B cells (Wong et al, 2002; Ghosh et al, 1999; Stoep et al, 2002; Holling et al, 2004). While putative candidate proteins have been suggested, these studies had largely been limited to EMSA assays probing for supershifts (Wong et al, 2002; Ghosh et al, 1999) and a genuine direct binder for the ARE elements has not been conclusively identified.

ZBTB48 (HKR3 or TZAP) was previously reported as a direct telomere-binding protein that acts as a negative regulator of telomere length (Jahn et al, 2017; Li et al, 2017; Kappei et al, 2017). In addition, it also binds to proximal promoters of a defined set of genes and moonlights as a transcriptional activator (Jahn et al, 2017). Here, we show that ZBTB48 binds directly to the two ARE sites within CIITA pIII. We demonstrate that by establishing open chromatin, ZBTB48 regulates inducible CIITA pIII expression and that constitutive B-cell-specific expression is affected in a ZBTB48 knockout mouse model, leading to a reduction in MHC-II-positive cells in primary B cells. Overall, we establish that ZBTB48 acts as a molecular on-/off-switch upstream of activating histone modifications and gene expression.

## Results

### ZBTB48 directly binds to two sites within CIITA pIII

In addition to its telomeric function, we have previously identified ZBTB48 as a transcriptional activator based on binding to proximal promoter regions and consequently lower transcript levels of the matched genes in ZBTB48 KO cells (Jahn et al, 2017). When revisiting our previous ChIP-seq data, we realized that ZBTB48 occupies CIITA pIII even in the absence of gene expression (Fig. 1A). Given the IFNγ-inducible nature of CIITA expression, we reasoned that CIITA might represent a previously unappreciated ZBTB48 target gene.

To identify the exact binding site(s) within the ChIP-seq peak, we performed in vitro reconstitution DNA pulldowns using nuclear protein extracts from U2OS cells using eight overlapping oligo probes, straddling the ZBTB48 ChIP-seq peak (Fig. 1B). In addition to the previously established binding to telomeric DNA (TTAGGG) (Jahn et al, 2017), three probes (2, 3, and 4) showed clear enrichment of ZBTB48, suggesting multiple binding sites within the CIITA pIII region (Fig. 1B,C). Therefore, we generated six smaller, 30 bp probes (a–f) and observed ZBTB48 enrichment on probes b and e. Subsequently, to narrow down the exact binding sites, we generated three overlapping 15 bp probes that span the sequences of probes b and e. Specific binding of ZBTB48 was identified to probes b.1 and e.2, that localize to −133 to −148 bp and −52 to −67 bp of CIITA pIII, respectively. Intriguingly, these binding sites correspond to both ARE-1 and ARE-2 (Fig. 1D), the critical regulatory elements of CIITA pIII expression.

### ZBTB48 interacts with CIITA pIII via its ZnF10, ZnF11, and C-terminal arm

We have previously demonstrated that among the 10 functional zinc fingers in ZBTB48, ZnF11 in combination with a short

C-terminal arm are necessary and sufficient for binding to telomeric DNA (Jahn et al, 2017; Zhao et al, 2018). To test whether binding at b.1 and e.2 is also mediated by ZnF11, we expressed point mutants of FLAG-ZBTB48 for each zinc finger and performed DNA pulldowns using the b.1 and e.2 probes (Fig. 1E). While FLAG-ZBTB48 WT and point mutants of ZnF1-9 efficiently bound to both b.1 and e.2, mutation of ZnF11 resulted in complete loss of its binding ability (Fig. 1F) in parallel to the expected loss in TTAGGG binding. In contrast, the ZnF10 mutant is capable of binding to telomeric DNA, but its binding to b.1 was strongly reduced and completely lost on e.2. We further confirmed the requirement of both ZnF10 and ZnF11 by using deletion constructs that lack ZnF1-9 and carry the same point mutations in either ZnF10 or ZnF11 (Fig. 1E). Indeed, FLAG-ZBTB48 ΔZnF1-9 was enriched on both probes, but a mutation in ZnF10 led to reduced enrichment on probe b.1 and loss of enrichment on e.2, while a mutation in ZnF11 led to a complete loss of binding on all probes (Fig. 1F). In addition, results from a previous co-crystal structure with telomeric DNA demonstrated that ZnF11 recognizes $G_4G_5G_6$ while two amino acid residues from the adjacent C-terminal arm, R611 and R614, are responsible for recognition of $T_2A_3$ (Zhao et al, 2018). To evaluate the involvement of the C-terminal arm, we tested R611A and R614A mutants in our pull-down assay (Fig. 1E). While the R611A mutant retained some enrichment on e.2 and R614A on telomeric DNA, neither mutant showed any binding to b.1. Furthermore, the R611A/R614A double mutant was incapable of binding to any of these sequences (Fig. 1F). These results demonstrate that ZBTB48 binds at two distinct sites within proximal CIITA pIII and requires ZnF10, ZnF11, and the C-terminal arm.

To further confirm the involvement of ZnF10 in CIITA binding, we generated a co-crystal structure of the e.2 probe with a construct containing ZnF10-11 and the C-terminal arm (Zhao et al, 2018) (Fig. 2A). Similar to telomeric DNA, base-specific interactions are conferred by ZnF11 in the major groove as well as by the C-terminal arm that turns back and lies across the phosphate backbone to fit in the minor groove (Fig. 2B; Table 1). Here, the RxxHxxR motif of ZnF11 recognizes $G_{10}G_{11}G_{12}$ with Arg589, His592, and Arg595 forming a bidentate hydrogen bond interaction with $G_{12}$, $G_{11}$, and $G_{10}$, respectively. This is equivalent to the interaction of both ZBTB48 and ZNF524 with telomeric repeats (Braun et al, 2023) and ZBTB10 with the telomeric variant repeat TTGGGG (Bluhm et al, 2018) (Fig. 2C). However, in contrast to the co-crystal structure with telomeric DNA (Zhao et al, 2018), ZnF10 does not lie outside of the CIITA e.2 DNA duplex but rather interacts with the phosphate backbone along the major groove (Fig. 2D). In the C-terminal arm, R614 forms hydrogen bonds with $G_{12}$ and on the complementary strand with $C_8T_9$ while R611 interacts with the phosphate backbone (Fig. 2A,E). In addition to establishing the molecular details underlying the interaction of the ZBTB48 with the CIITA pIII sequence, these results establish the potential separation of function mutations for ZBTB48's role at telomeres and promoters—at least for binding to CIITA pIII.

### ZBTB48 mediates IFN γ-induced expression of CIITA

The overlap of ZBTB48 binding sites with the ARE elements suggests a regulatory role of ZBTB48 in CIITA pIII expression. To

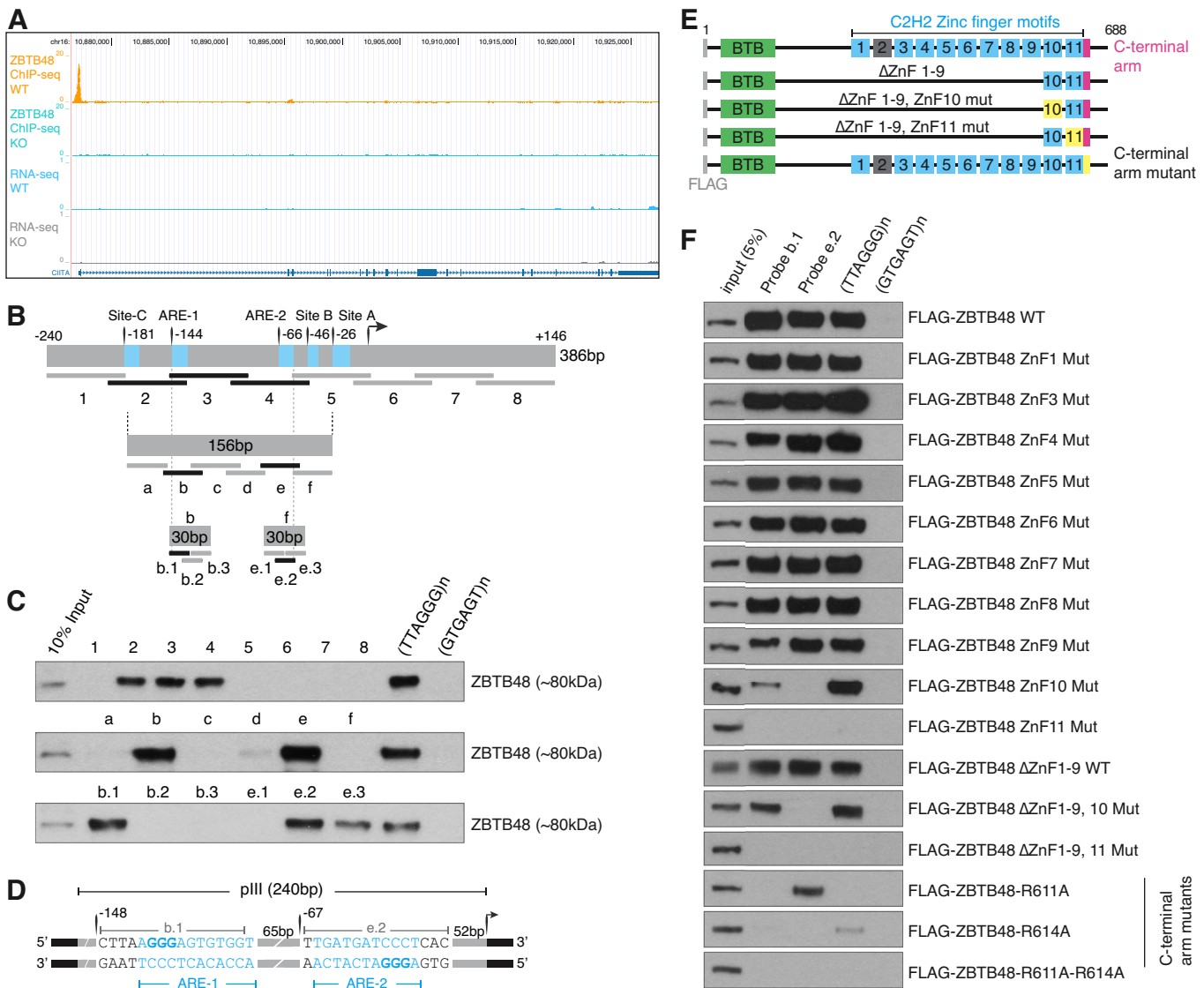

**Figure 1. ZBTB48 directly binds to CIITA pIII.**

(A) ChIP-seq tracks depicting ZBTB48 binding peaks at CIITA pIII in U2OS cells but not in the KO cells ($n = 4$). No reads are observed in the corresponding RNA-seq tracks in these uninduced cells ($n = 5$; data re-analyzed from Jahn et al (Jahn et al, 2017)). (B) Schematic of DNA probe design encompassing the binding peak that was used in the DNA pull-down assay. Transcription start site (TSS) and known regulatory elements are indicated and binding in the pulldowns is denoted by probes in darker shade. (C) Western blot of DNA pull-down assay using U2OS nuclear extracts demonstrates ZBTB48 binding at two sites within pIII. Telomeric (TTAGGG) and scramble control (GTGAGT) sequence are used as positive and negative controls, respectively. (D) Schematic of ZBTB48 binding sites within CIITA pIII with their relative distances to the TSS and their sequence overlap to the originally mapped ARE elements (Ghosh et al, 1999). (E) Schematic of FLAG-tagged ZBTB48, a 688-amino acid protein, containing N-terminal BTB domain (green), 11 zinc fingers (ZnFs) and a C-terminal arm (pink). The functional ZnFs are in blue and degenerate ZnF2 is in gray. Deletion constructs lacking ZnF1-9 with and without points mutations in either ZnF10 or ZnF11 and WT ZBTB48 with mutations in C-terminal arm are depicted. (F) DNA pull-downs with CIITA pIII probes b.1, e.2, telomeric (TTAGGG) and control sequence (GTGAGT) for FLAG-ZBTB48 WT, ZnF mutants and C-terminal arm mutants according to (D). Source data are available online for this figure.

investigate this, we compared CIITA expression in each of five U2OS WT and ZBTB48 KO clones as biological replicates upon IFNγ treatment. Using primers that detect all CIITA isoforms (pan-CIITA), we observed only very low baseline expression of CIITA in untreated cells (Fig. 3A). In contrast, IFNγ stimulation resulted in a strong, >1000-fold induction of CIITA in WT cells with a 34-fold higher expression than in the ZBTB48 KO clones. Although IFNγ treatment induced expression of all three CIITA promoters, no significant difference in pI expression was observed between WT and KO clones and, overall, pI induction was modest. pIII was the most strongly induced isoform and its expression was 154-fold higher in WT compared to KO clones (Fig. 3A). Similarly, transcript levels of pIV were also on average 17-fold higher in WT clones. The loss of pIII expression was further recapitulated by RNA-seq data from the same clones, where the enrichment of transcripts from the first exon that is unique to pIII is absent in the

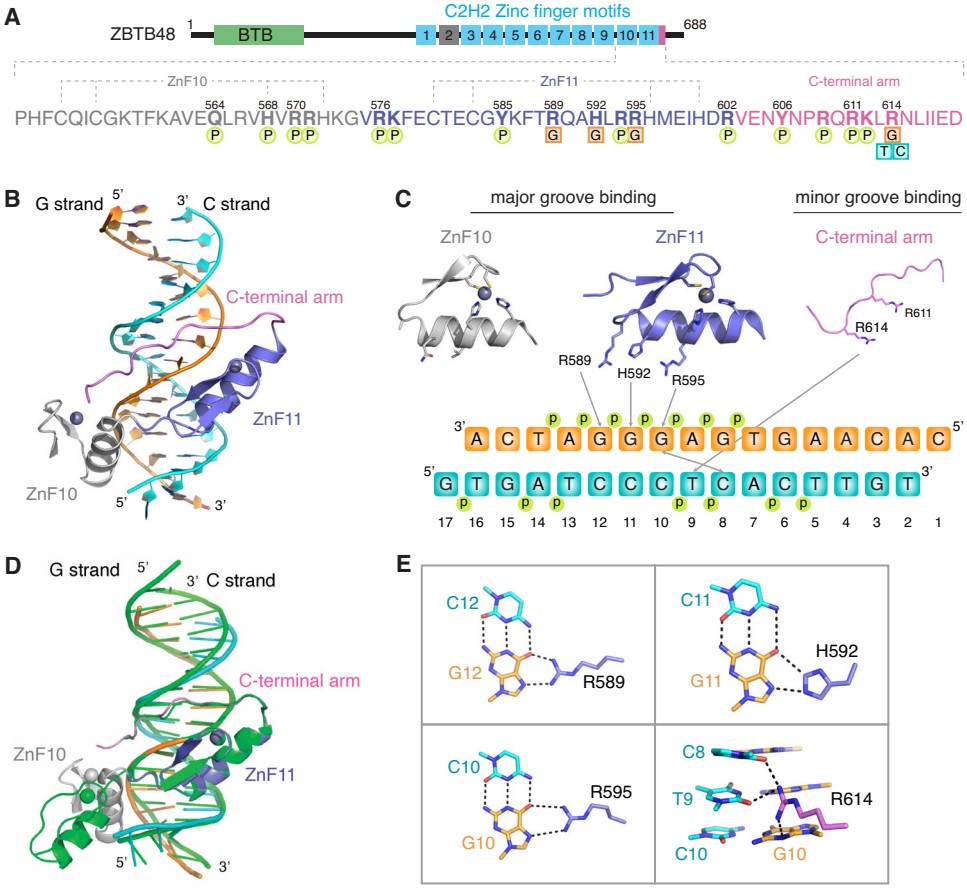

**Figure 2.  ZBTB48 binding to CIITA pIII requires ZnF10 and ZnF11.**

(A) Schematic representation of the ZBTB48 protein structure and a zoom in on the expression construct containing ZnF10, ZnF11 and the C-terminal arm. Base-specific interactions (squares) and interactions with the DNA phosphate backbone (circles) are indicated. (B) The overall co-crystal structure of the essential ZBTB48 DNA-binding domains with the e.2-binding site within CIITA pIII. (C) Base-specific interactions depicted for each of the three ZBTB48 domains involved in binding to CIITA pIII. Binding of ZnF11 to the GGG sequence involves its RxxHxxR motif. (D) Superposition of the co-crystal structure of the essential ZBTB48 DNA-binding domains with telomeric DNA (bright green) and the e.2-binding site within CIITA pIII. Note how ZnF10 contributes to the interaction with the e.2 sequence while lies outside the DNA duplex in case of the telomeric DNA. (E) Details of base-specific interactions for ZnF11 and the C-terminal arm. Hydrogen bonds are depicted as dashed lines.

KO after IFNγ stimulation (Fig. 3B). ZBTB48 binding to CIITA pIII and loss of pIII mRNA transcript levels in ZBTB48 KO clones was further confirmed in HeLa cells (Fig. EV1A,B). Overall, CIITA transcript levels were abundant in the WT clones with only minimal levels detected in the KO clones, clearly demonstrating that ZBTB48 is critical for IFNγ-induced CIITA pIII expression.

To validate the downstream consequence of reduced CIITA expression in ZBTB48 KO cells, we analyzed transcript levels of HLA-DPA, -DQB1, -DMB, -DMA, -DRB5, and CD74 as representative MHC-II genes. Again, upon IFNγ stimulation, U2OS WT clones expressed 15- to 29-fold higher levels of the HLA genes compared to ZBTB48 KO clones (Fig. 3C). To globally interrogate this effect, we analyzed expression differences in our RNA-seq data comparing the WT and KO clones with or without IFNγ. Overall, a strong induction of interferon-stimulated genes (ISGs) was observed in both WT and KO clones upon IFNγ stimulation (Fig. EV1C,D; Dataset EV1). In agreement with our qPCR data, expression levels of CIITA, HLA-DQB1, -DMB and -DMA were significantly higher in the IFNγ treated WT clones based on cut-

offs of fold change >2 and adjusted *p* value < 0.01 (Fig. 3D). The other MHC-II genes were detected but did not achieve significance due to the stringent cut-offs, but overall the expression of the entire gene family was elevated in WT clones (Fig. EV1E,F). Consistent with our previous data (Jahn et al, 2017), expression of MTFP1, VWA5A, and PXMP2, were also decreased in ZBTB48 KO clones irrespective of IFNγ treatment (Figs. 3D and EV1G). Strikingly, beyond MHC-II-related transcripts, no other ISGs were differentially expressed upon IFNγ stimulation between WT and KO clones (Fig. 3D). These results suggest that the impact of ZBTB48 in response to IFNγ stimulation is exclusive to CIITA expression and that ZBTB48 itself likely does not respond to IFNγ, even though at present we cannot exclude the possibility of IFNγ-induced post-translational modifications contributing to the ZBTB48-dependent CIITA activation.

We further validated the lower expression of CIITA and its target genes in IFNγ-treated U2OS ZBTB48 KO clones at the protein level by Western blot. In agreement with the qPCR data, CIITA, HLA-DMB, and CD74 protein levels were low to

**Table 1.  Co-crystal structure data collection and refinement statistics.**

|  | ZBTB48-CIITA DNA |
|---|---|
| Wavelength (Å) | 0.979 |
| Space group | $P4_32_12$ |
| Cell parameters |  |
| a, b, c (Å) | 126.56, 46.86, 79.05 |
| α, β, γ (°) | 90, 124.89, 90 |
| Resolution (Å) | 40.00–2.90 (2.95–2.90) |
| $R_{merge}$ (%) | 11.7 (86.7) |
| $CC_{1/2}$ (%) | 98.9 (81.5) |
| $I/σI$ | 14.6 (1.0) |
| Completeness (%) | 98.9 (91.4) |
| Average redundancy | 6.2 (5.0) |
| **Refinement** |  |
| No. of reflections (overall) | 7083 |
| No. of reflections (test set) | 330 |
| $R_{work}/R_{free}$(%) | 23.34/29.12 |
| Protein | 1172 |
| DNA | 1300 |
| ZN | 4 |
| B factors (Å²) |  |
| Protein | 46.44 |
| DNA | 69.02 |
| ZN | 30.58 |
| R.m.s. deviations |  |
| Bond lengths (Å) | 0.003 |
| Bond angles (°) | 0.473 |
| RAMPAGE Ramachandran plot % residues |  |
| Favored | 92.42 |
| Allowed | 7.58 |
| Outliers | 0 |

undetectable across the five KO clones (Fig. 3E). To further validate this, we measured global protein expression using label-free quantitative mass spectrometry analysis in all five WT and KO clones treated with or without IFNγ. Here, we detected and quantified HLA-DRA, -DRB1, -DPA1, -DRB4, DMB, and CD74 as members of the CIITA-MHC-II axis. As for mRNA levels, ISGs were induced in both WT and KO clones upon IFNγ treatment, but only the MHC-II proteins showed significantly higher expression in WT clones (Figs. 3F and EV2A–E; Dataset EV2). Once more, the changes observed between WT and KO remained limited to factors that were previously described to differ independent of IFNγ-induction and the CIITA-MHC-II axis. The combined transcriptomic and proteomic profiling validates that in the cascade of cellular IFNγ response, ZBTB48 specifically enables induction of CIITA expression—primarily its pIII transcript—, which in return drives MHC-II expression.

## ZBTB48 loss results in diminished MHC-II expression in vivo

CIITA pIII is conserved in mice including the triple G motifs that are recognized by ZBTB48 through base-specific contacts within the b.1 and e.2 sequences (Fig. 4A). Likewise, the critical domains in ZBTB48 have 100% protein sequence identified between the human and mouse orthologs (Fig. EV3A), suggesting that ZBTB48 might confer similar regulation in mice. To test the in vivo effects of loss of ZBTB48 on MHC II, we established a ZBTB48 KO mouse model by deleting the ATG-containing exon 2 (Figs. 4B,C and EV3B). ZBTB48$^{-/-}$ (KO) and ZBTB48$^{+/-}$ (Het) mice were born at Mendelian ratios and did not display any obvious growth defects compared to their wild-type (WT) littermates (Fig. EV3C). However, female but not male KO mice presented with mild splenomegaly in 11–14 week-old animals (Fig. 4D,E) with no effect on total body weight (Fig. EV3D). Given that spleen is a B-cell-rich organ, we next interrogated molecular phenotypes by assessing MHC-II levels. Lineage-specific cells were isolated from peripheral blood, thymus, spleen as well as bone marrow (BM) and MHC-II expression was quantified by fluorescence-activated cell sorting (FACS). While the relative abundance of different cell types remained unchanged in both spleen and bone marrow (Fig. EV3E,F), the percentage of MHC-II-positive B cells was reduced in ZBTB48 KO mice. Splenic B cells presented with a moderate loss in MHC II-positive cells in the KO (69%) mice compared to both WT (85%) and Het (89%) animals (Fig. 4F) in contrast to a more pronounced loss in the BM with approximately three times less MHC-II-positive B cells in the ZBTB48 KO (28%) mice as compared to that in WT (72%) mice (Fig. 4G). In addition, in BM cells a dose-dependent effect was detectable with a significant, intermediary reduction of MHC-II-positive B cells in Het animals. Given the more pronounced reduction in BM but a milder reduction of MHC-II-positive B cells in peripheral blood—90% in the KO vs 98% in the WT mice (Fig. EV3G,H)—we wondered whether MHC-II loss originates during B-cell development. Here, the hematopoietic stem and progenitor cells (HSPCs) in the BM were comparable across all genotypes and did not display any significant change in the number of the relatively moderate levels of MHC-II-positive cells (Fig. EV4A,B). In contrast, developmental B-cell populations, but not thymic developmental T-cell populations (Fig. EV4C,D), were depleted for MHC-II-positive cells without affecting the relative abundance of cell populations belong to different B-cell differentiation states across genotypes (Fig. EV4E,F). In the early pro-B stage A, the number of MHC-II-positive B cells reduced by more than half in KO mice (17%) compared to WT (37%) and Het (34%) littermates (Fig. 4H). A similar reduction was seen in pro-B stage B cells while pro-B stage C cells did not present a significant reduction in MHC-II levels. Remarkably, the loss of MHC-II-positive cells peaked in the pre-B (stage D) subpopulation, where MHC-II-positive cells were barely detected in KO mice (3.7%), compared to 46% and 28.8% MHC-II-positive cells in WT and Het mice, respectively (Figs. 4H,I and EV4G,H). As cells developed further, this difference tapered off, with on average five times more MHC-II-positive cells in WT (69%) and four times more in Het (52%) compared to KO mice (13%) in the immature (E) subpopulation (Fig. 4H). Finally, mature B cells (F subpopulation) in WT animals

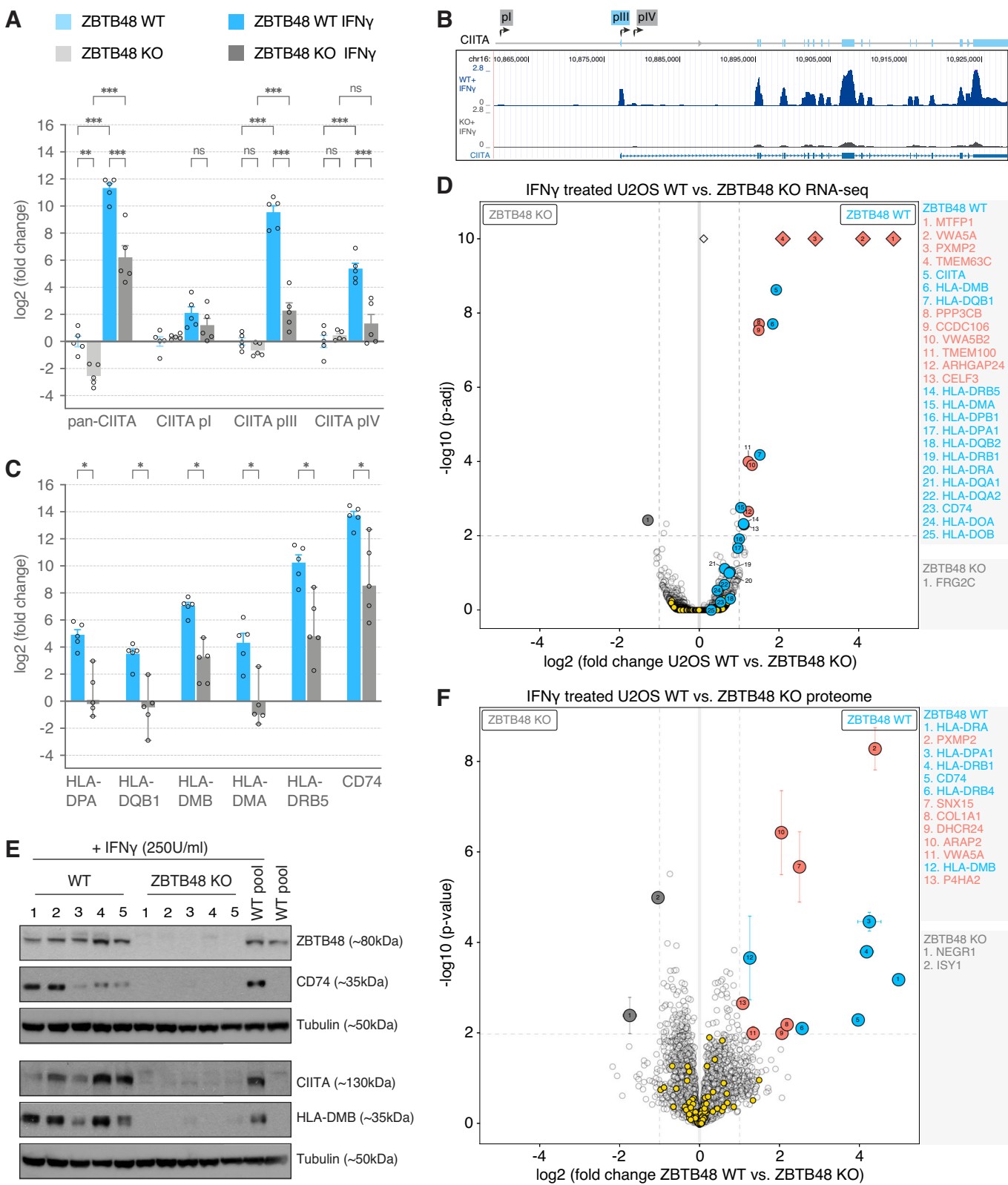

◀ **Figure 3.   ZBTB48 is required for IFNγ-induced expression of CIITA pIII.**

(A) Relative mRNA expression of total (pan-)CIITA and the three promoter-specific transcripts in each of five independent U2OS WT and ZBTB48 KO clones treated with or without 250 U/ml IFNγ for 24 h. Data represents mean ± SEM. The $\log_2$ fold change is calculated relative to the average of five untreated WT clones. $p$ values were calculated by two-way ANOVA ($n = 5$); **$p < 0.01$, ***$p < 0.001$. pan-CIITA: WT vs KO: $p = 0.002$; WT vs WT + IFNγ: $p < 1E-15$; KO vs KO + IFNγ: $p < 1E-15$, WT + IFNγ vs KO + IFNγ: $p = 8.068E-10$, for CIITA pI: WT + IFNγ vs KO + IFNγ: $p = 0.695$, for CIITA pIII: WT vs KO: $p = 0.930$; WT vs WT + IFNγ: $p < 1E-15$; KO vs KO + IFNγ: $p = 3.89E-04$, WT + IFNγ vs KO + IFNγ: $p = 2E-15$, for CIITA pIV: WT vs KO: $p = 0.991$; WT vs WT + IFNγ: $p = 2.079E-10$; KO vs KO + IFNγ: $p = 0.734$, WT + IFNγ vs KO + IFNγ: $p = 4.633E-07$. (B) RNA-seq tracks at the CIITA gene in five IFNγ-induced U2OS WT and ZBTB48 KO clones ($n = 5$). The positions of the three promoters are indicated in the schematic above. (C) mRNA expression changes of HLA genes in the five U2OS WT and ZBTB48 KO clones treated with 250 U/ml IFNγ for 24 h. Data represents mean ± SEM. The $\log_2$ fold change is calculated relative to the average of five untreated WT clones. $p$ values were calculated by multiple $t$ test controlling the FDR by two-stage step-up method of Benjamini, Krieger and Yekutieli to correct for multiple comparisons ($n = 5$); *$p < 0.05$. The $p$ values for the WT + IFNγ vs KO + IFNγ comparisons are as follows: HLA-DPA: $p = 0.003$, HLA-DQB1: $p = 0.004$, HLA-DMB: $p = 0.002$, HLA-DMA: $p = 0.004$, HLA-DRB5: $p = 0.005$, CD74: $p = 0.007$. (D) Differential expression analysis of the RNA-seq quantification, comparing each of five U2OS WT and ZBTB48 KO clones for U2OS. DESeq2 was used for differential expression analysis and log fold changes were shrunk using the original DESeq2 shrinkage estimator (normal). (E) Western blot for CIITA, HLA-DMB and CD74 protein expression in five U2OS ZBTB48 KO clones compared to five WT clones each when treated with 250 U/ml IFNγ for 48 h. The parental U2OS WT pool treated with or without IFNγ treatment is included for reference. (F) Protein expression analysis comparing five U2OS WT and ZBTB48 KO clones treated with IFNγ (250 U/ml, 48 h) by label-free quantitative mass spectrometry. Two-dimensional error bars represent the standard deviation based on iterative imputation cycles during the label-free analysis to substitute missing values (e.g., no detection in the KO clones) and the measure of center is mean. $p$ values were calculated by independent sample $t$ test ($n = 5$). For volcano plots in (D, F), specifically enriched hits are distinguished from background proteins by a two-dimensional cut-off of >twofold enrichment and $p < 0.01$. The hits belonging to the CIITA-MHC-II family are shown in blue and the rest of the IFNγ response genes (ISGs) in yellow, ZBTB48 targets independent of IFNγ in salmon and hits enriched in KO clones are shown in gray. Source data are available online for this figure.

are nearly all MHC-II-positive while KO mice had ~20% less MHC-II-positive cells, similar to the frequency observed in splenic B cells. Of note, CIITA expression in WT mice is reflected by a sustained high CIITA pIII expression with concomitant stable ZBTB48 mRNA expression levels (Fig. EV4I). In sum, we observed a ZBTB48 dose-dependent reduction in the BM-derived MHC II expressing pro- and pre-B cells, likely due to transcriptional silencing of CIITA pIII. Overall, these data clearly illustrate that ZBTB48 is a critical regulator of B-cell-specific MHC-II expression in vivo.

## ZBTB48 promotes open chromatin at CIITA pIII

Constitutive binding of ZBTB48 at CIITA pIII, even when the promoter is inactive (Figs. 1A and EV1A), suggests that ZBTB48 primes transcriptional activation. To confirm that ZBTB48 binds independently of IFNγ, we performed ChIP-qPCR comparing each of four U2OS WT and ZBTB48 KO clones both with and without IFNγ-treatment. In contrast to no detectable enrichment at the upstream CIITA pI, ZBTB48 binds strongly at CIITA pIII in the WT clones with >450-fold enrichment relative to the ZBTB48 KO clones (Fig. 5A). In agreement with constitutive binding, this interaction was not altered upon IFNγ induction. Similar results were obtained for the MTFP1 promoter, which is regulated by ZBTB48 irrespective of IFNγ (Jahn et al, 2017). Interestingly, we detected an additional relatively more moderate 17-fold enrichment of ZBTB48 compared to the KO clones about 6–7 kb upstream of pIII. Likewise, an eightfold enrichment was also detected at pIV (Fig. 5A). These comparably weak interactions might represent remnants of chromatin looping that contribute to CIITA regulation. Again, ZBTB48 binding at various loci within the CIITA locus is not affected by IFNγ, further validating that ZBTB48 itself does not respond to IFNγ.

Transcriptional programs primed for rapid induction are marked by pre-existing activating histone marks and accessible chromatin at promoters and enhancers. To ascertain ZBTB48-dependent reshaping of the promoter landscape, we compared the activating H3K4me3 histone mark (Lauberth et al, 2013) at the CIITA promoter region in four U2OS WT and ZBTB48 KO clones each by ChIP-qPCR. Indeed, IFNγ treatment induced H3K4me3

marks at both CIITA pIII and pIV in WT but not KO clones (Fig. 5B). Again, using MTFP1 as a constitutively expressed gene regulated by ZBTB48, H3K4me3 marks were enriched at the MTFP1 promoter in the WT clones independent of IFNγ treatment (Fig. 5B). We obtained similar results in SUDHL4 cells, a diffuse large B-cell lymphoma (DLBCL) cell line, where CIITA pIII is constitutively expressed (Fig. EV5A–C). To test whether H3K4me3 is altered at promoters of other ZBTB48 target genes, we performed H3K4me3 ChIP-seq in three U2OS WT and ZBTB48 KO clones treated with IFNγ. Again, enrichment of H3K4me4 over CIITA pIII was observed in the WT clones (Fig. 5C; Dataset EV3). In addition, the H3K4me3 enrichment in WT cells extends to pIV, indicative of an active promoter status across the entire region. Concomitantly, downstream of CIITA induction, the majority of genes in the HLA cluster displayed robust H3K4me3 signal in WT but not KO clones (Fig. EV5D; Dataset EV3). However, not all promoters regulated by ZBTB48 depicted a similar loss of H3K4me3 levels in KO clones. For instance, the ZBTB48 target gene SNX15 retained its H3K4me3 signature at the proximal promoter in KO clones despite a loss of gene expression (Fig. EV5E). When globally aligning the ZBTB48 and H3K4me3 ChIP-seq data, the majority of ZBTB48 binding sites show strong concordance with H3K4me3 signal and the overall profiles remain constant between WT and KO clones (Fig. 5D; Dataset EV3). Since only a subset of ZBTB48 binding sites respond to ZBTB48 deletion with a H3K4me3 reduction, ZBTB48 likely does not influence this histone modification directly. This is in agreement with the absence of H3K4me3 at CIITA pIII in uninduced U2OS WT cells despite the constitutive binding of ZBTB48 to the promoter.

The above results suggest that ZBTB48 might act on upstream chromatin regulation. To investigate a putative effect on chromatin accessibility, we performed Formaldehyde-assisted isolation of regulatory elements (FAIRE) (Rodríguez-Gil et al, 2018) followed by qPCR in four U2OS WT and ZBTB48 KO clones. Chromatin accessibility at CIITA pIII was 6-fold higher in the WT clones, and this difference increased to 24-fold upon IFNγ treatment (Fig. 6A). In comparison, the constitutively active MTFP1 promoter also displayed open chromatin in WT but not KO clones and as expected this did not significantly alter upon

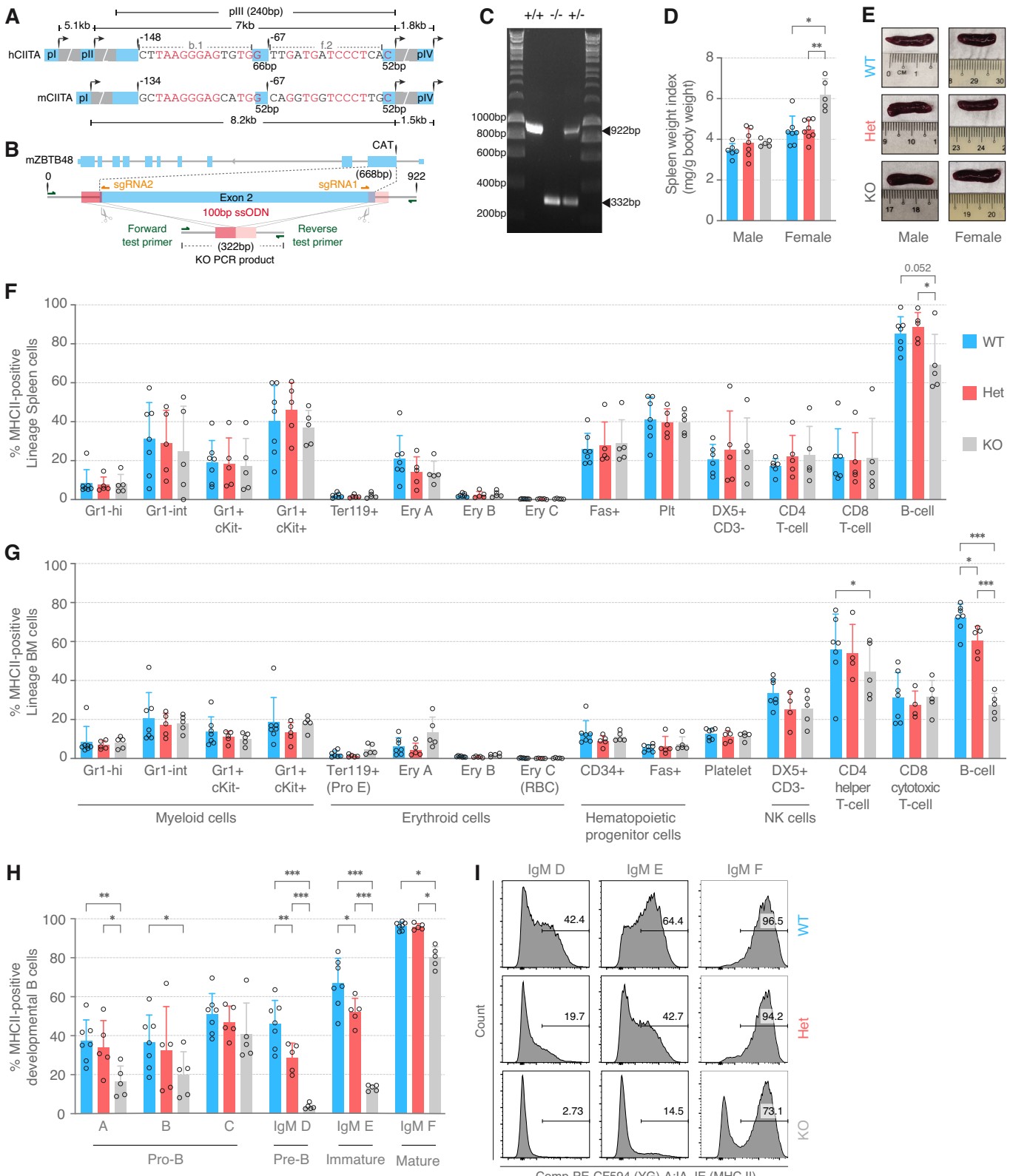

◄   **Figure 4.   Loss of ZBTB48 diminishes B cell-specific MHC-II expression in vivo.**

(A) Schematic of the CIITA promoter locus depicting all four promoters in human and three in mice. The sequence conservation between human and mice pIII of b.1 and f.2 is indicated. (B) Schematic overview of sgRNAs and ssODN for CRISPR/Cas9 mediated deletion at exon 2 of the mZBTB48 locus. The ssODN containing 50 bp homology arms flanking the cut sites is depicted. The primers (green) used for genotype confirmation flanking the ssODN and the resulting PCR product size is indicated. (C) Mouse genotyping results using primers indicated in (B) and DNA obtained from tail clippings. PCR products from ZBTB48$^{-/-}$ (KO) and ZBTB48$^{+/-}$ (Het) mice were separated on a 1% agarose gel to differentiate between WT (922 bp) and KO (322 bp). (D) Spleen weight index (weight of spleen in mg/body weight in g) calculated separately for males and females across the three genotypes for 11–14 weeks-old mice. Data represents mean ± SD. $p$ values were calculated by Mann-Whitney test (male: WT $n = 6$, Het $n = 7$, KO $n = 5$; female: WT $n = 7$, Het $n = 8$, KO $n = 5$); *$p < 0.05$, **$p < 0.01$. For female spleen weight index comparisons: WT vs KO: $p = 0.020$, Het vs KO: $p = 0.002$. (E) Representative spleen images comparing spleen size across the three genotypes in males and females. (F–H) Percentage of MHC-II-positive lineage-specific cells in the spleen (F), bone marrow (BM) (G), and developmental B cells (H) in WT ($n = 7$), Het ($n = 5$) and KO ($n = 5$) mice. Gr1-hi, Gr1 high; Gr1-int, Gr1 intermediate. Data represents mean ± SD. $p$ values were calculated by two-way ANOVA with Sidak correction for multiple comparisons; *$p < 0.05$, **$p < 0.01$, ***$p < 0.001$. $p$ values for the percentage MHC-II cells comparisons are as follows: B cells in spleen: Het vs KO: $p = 0.02$; B cells in BM: WT vs Het: $p = 0.018$, WT vs KO: $p < 1E-15$, Het vs KO: $p = 1.657E-11$; CD4 T cells in BM: WT vs KO: $p = 0.022$; Developmental B cells: stage A: WT vs KO: $p = 0.001$, Het vs KO: $p = 0.013$; stage B: WT vs KO: $p = 0.012$; D (pre-B): WT vs Het: $p = 0.008$, WT vs KO: $p = 4.124E-11$, Het vs KO: $p = 2.249E-04$; E (immature): WT vs Het: $p = 0.031$, WT vs KO: $p = 3.400E-14$, Het vs KO: $p = 1.040E-08$; F (mature): WT vs KO: $p = 0.016$, Het vs KO: $p = 0.032$. The developmental stage of the B cells when gated by IgM in (H) are indicated below. (I) Representative flow cytometry analysis of MHC-II-positive cells in D (pre-B), E (immature), and F (mature) subpopulations from WT, Het, and KO littermates when gated by IgM. Source data are available online for this figure.

IFNγ stimulation. CIITA pI, pIV, and the 6–7 kb upstream site were unaffected by the loss of ZBTB48, demonstrating that ZBTB48 specifically determines the chromatin state precisely at its binding site within CIITA pIII. To further establish this function of ZBTB48, we performed genome-wide FAIRE-seq analysis of open chromatin after IFNγ induction in U2OS WT and ZBTB48 KO clones. Again, the ZBTB48 binding peak at CIITA pIII coincided with the location of opened chromatin (Fig. 6B; Dataset EV4). Similar results were observed at other ZBTB48-dependent genes such as MTFP1 and SNX15 (Fig. EV5F,G), suggesting that ZBTB48 in general ensures chromatin accessibility at the promoters of its targets.

Based on these results, we propose that ZBTB48 functions as a priming factor that sets the stage for promoter activation (Fig. 6C). In such a model, ZBTB48 binding at nucleosome-compacted promoters induces a nucleosome-depleted region to subsequently grant access to other transcription factors and chromatin remodeling complexes. In the case of non-APCs, this requires additional IFNγ stimulation and binding by IFNγ-dependent transcription factors while corresponding factors are likely constitutively expressed in B cells. In both scenarios, this will subsequently solidify the open chromatin state and trigger transcription initiation by RNA polymerase II.

## Discussion

The precisely fine-tuned and strictly regulated cell type-specific expression of MHC II is primarily regulated at the transcriptional level by the differential usage of the three independent promoters of CIITA. In addition to being IFNγ-inducible, pIII predominantly drives the constitutive expression of CIITA in B cells. Here, we describe ZBTB48 as a key regulator of both IFNγ-inducible and constitutive in vivo expression of CIITA pIII. Prior work had established ARE-1 and ARE-2 as the essential regulatory elements for CIITA pIII (Wong et al, 2002; Ghosh et al, 1999). Intriguingly, for at least ARE-2, the authors had also implicated a GGG motif as critical for expression regulation (Ghosh et al, 1999). Our serendipitous discovery of ZBTB48 hence ideally fits in this picture with ZBTB48's ability to recognize GGG motifs via ZnF11. While other ARE-binding proteins, including AML2/3, CREB1 and

ATF1, had been previously suggested (Wong et al, 2002; Ghosh et al, 1999; Stoep et al, 2002; Holling et al, 2004), these data are limited to supershifts in EMSA assays and therefore did not include in vivo validation. It remains to be evaluated whether any of these proteins specifically compete with ZBTB48 for binding to the ARE elements. Nonetheless, the near complete absence of CIITA pIII induction in both U2OS and HeLa ZBTB48 KO clones suggests that at least in these cellular contexts ZBTB48 is the essential ARE transducer.

While we observed similarly striking phenotypes in pre-B cells in our ZBTB48 KO mouse model, the MHC-II loss was less pronounced in mature B cells as well as in earlier and later B-cell developmental stages. These data suggest that there is heterogeneity in the B-cell population either in their CIITA promoter choice and/or in terms of expression of potentially redundant regulators. Even in systemic CIITA knockout mice, 1–3% MHC-II-positive splenic B cells were detected (Williams et al, 1998), implying that the small portion of ZBTB48-independent, MHC-II-positive pre-B cells might be selected for and populate the mature B-cell stages. While at present we cannot explain why only specifically female ZBTB48$^{-/-}$ mice present with a mild splenomegaly, quality control mechanisms triaging out MHC-II-negative precursors, might contribute to the observed phenotype. Such a scenario should ultimately lead to stem cell exhaustion in the B-cell lineage with increasingly smaller numbers of MHC-positive cells in mature B cell populations and/or a reduction in the corresponding cell numbers. However, in contrast to the above, systemic CIITA knock-outs overall lack cell surface MHC-II expression on splenic B cells and DCs, and their macrophages and somatic tissue cells also do not express MHC II upon IFNγ induction (Chang et al, 1996; Williams et al, 1998). Since studies in mice targeting exclusively pIII are lacking, a direct comparison with our results is impossible, but our data imply that exclusive loss of pIII-driven CIITA can be partially compensated at least in young animals.

Molecularly, our results demonstrate that ZBTB48 acts as a priming factor that is essential to activate its target genes. Intriguingly, the two ZBTB48 binding sites are ~80 nucleotides apart from each other (Fig. 1D), which in a nucleosome context places both binding sites in adjacent positions (Luger et al, 1997). Given that many ZBTB family members are known to homo- or

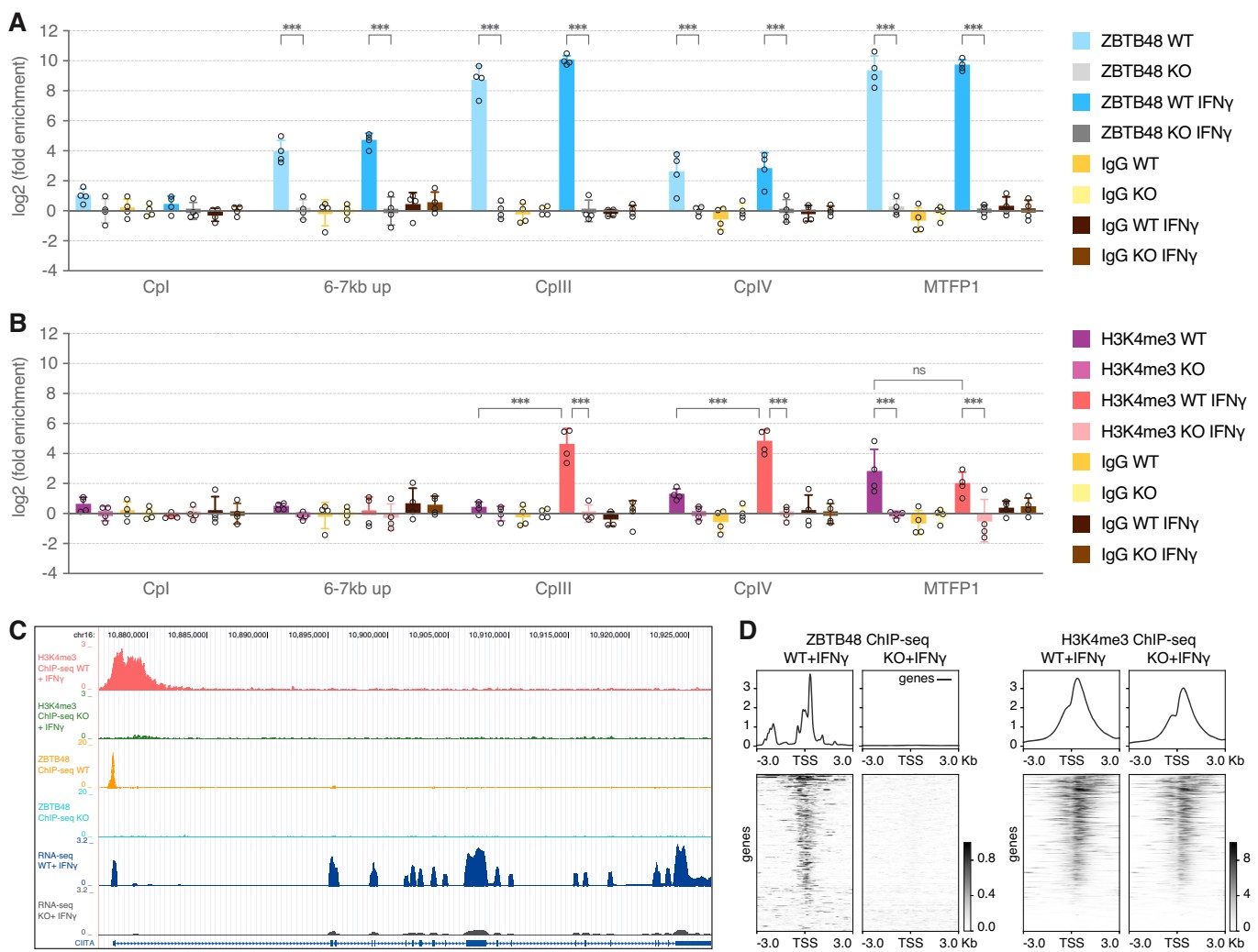

**Figure 5. ZBTB48 binding is required for H3K4me3 deposition at CIITA pIII.**

(A) ChIP reactions from four independent U2OS WT and ZBTB48 KO clones analyzed by qPCR for CIITA pI, pII (6–7 kb up), pIII, pIV. MTFP1 promoter is used as a positive control. The data was normalized to a gene desert region and fold change was calculated relative to the average of WT clones. Data represents mean ± SD. $p$ values were calculated by two-way ANOVA with Sidak correction for multiple comparisons ($n = 4$); *$p < 0.05$, **$p < 0.01$, ***$p < 0.001$. $p$ values for ZBTB48 enrichment comparisons are: 6–7 kb up: WT vs KO: $p = 6.505E-12$, WT + IFNγ vs KO + IFNγ: $p < 1E-15$; CpIII: WT vs KO: $p < 1E-15$, WT + IFNγ vs KO + IFNγ: $p < 1E-15$; CpIV: WT vs KO: $p = 4.481E-06$, WT + IFNγ vs KO + IFNγ: $p = 4.977E-07$; MTFP1: WT vs KO: $p < 1E-15$, WT + IFNγ vs KO + IFNγ: $p < 1E-15$. (B) H3K4me3 ChIP reactions from four independent U2OS WT and ZBTB48 KO clones analyzed by qPCR as in (A). $p$ values for H3K4me3 enrichment comparisons are: CpIII: WT vs WT + IFNγ: $p = 4.637E-12$, WT + IFNγ vs KO + IFNγ: $p = 6.400E-14$; CpIV: WT vs WT + IFNγ: $p = 4.219E-09$, WT + IFNγ vs KO + IFNγ: $p = 7E-15$; MTFP1: WT vs WT + IFNγ: $p = 0.967$, WT vs KO: $p = 2.894E-06$, WT + IFNγ vs KO + IFNγ: $p = 9.913E-05$. (C) ChIP-seq tracks depicting ZBTB48 and H3K4me3 binding peaks at CIITA pIII in U2OS WT cells but not in the KO cells. The H3K4me3 ChIP-seq tracks are overlays of three IFNγ-induced WT and ZBTB48 KO clones. RNA-seq tracks at CIITA in IFNγ-induced five WT and ZBTB48 KO U2OS clones. The ZBTB48 ChIP-seq tracks are the same as described in Fig. 1A. (D) ChIP-seq heatmap at the top 500 ZBTB48 binding sites (Jahn et al, 2017) (ranked by the mean of count per million (CPM) reads (Ramírez et al, 2016)) for co-occupancy of the H3K4me3 signal. Plots are centered on the TSS ± 3 kb. Source data are available online for this figure.

heterodimerize via their BTB domain (Piepoli et al, 2022), it is tempting to speculate that recognition of nucleosome-bound CIITA pIII is achieved by a ZBTB48 dimer. Upon opening of the locus, CIITA pIII would subsequently become accessible for other transcription factors that in turn are required to trigger gene expression (Fig. 6C). In the case of B cells these factors are likely constitutively expressed while in non-APCs their activity requires IFNγ stimulation in addition to the pre-existing ZBTB48-dependent CIITA pIII priming. It will be important in future work to establish how ZBTB48 promotes histone

displacement, how it may work in concert with other factors and how CIITA pIII regulation may spillover to other CIITA promoters as seen with CIITA pIV in U2OS cells (Fig. 3A). Meanwhile, ZBTB48 could be equated to an on-off light switch while additional transcription factors function as dimmer switches. Hence, while ZBTB48 determines whether gene activation is possible at all, the scale of transcriptional output is regulated downstream of the priming activity by other factors whose own activity limits CIITA pIII expression to B cells and non-APCs upon IFNγ induction.

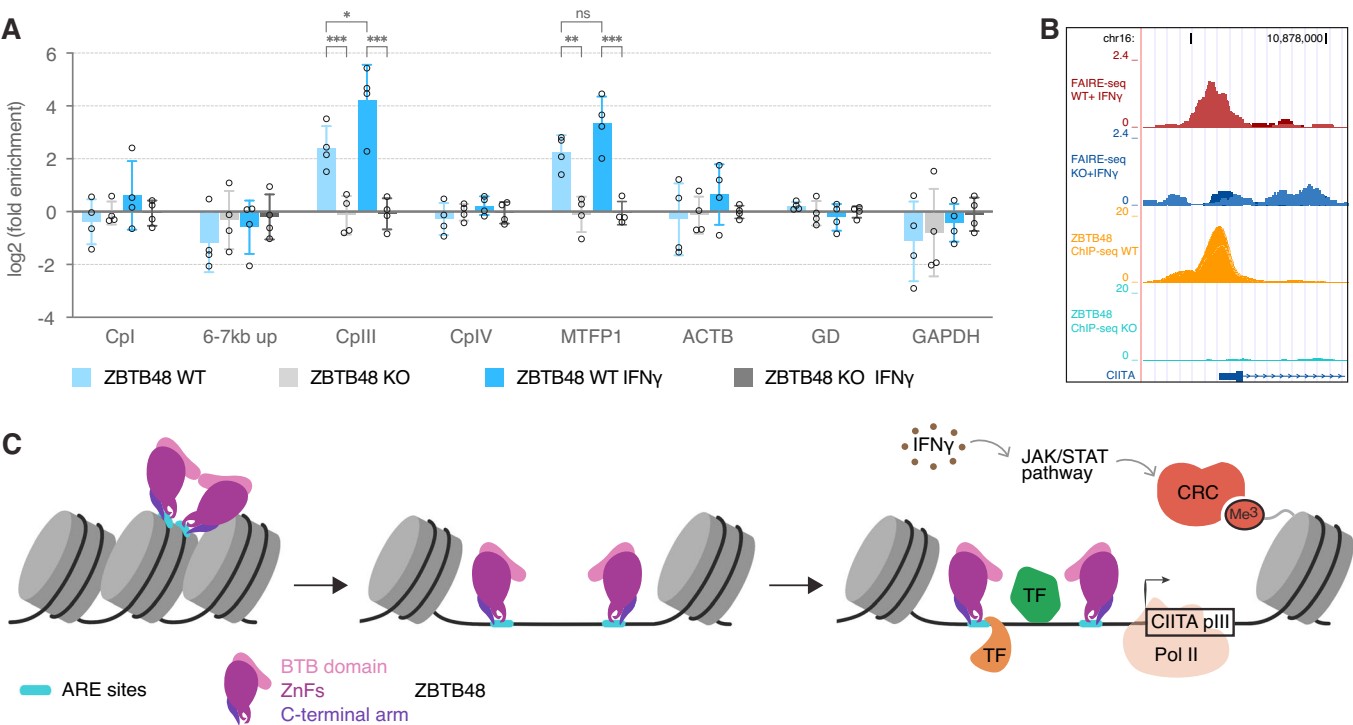

**Figure 6.   ZBTB48 establishes open chromatin at CIITA pIII.**

(**A**) FAIRE reactions from four independent U2OS WT and ZBTB48 KO clones with and without IFNγ induction analyzed by qPCR for CIITA pI, pII (6–7 kb up), pIII, pIV. MTFP1, GAPDH, and ACTB (beta actin) promoters are used as a positive controls. Gene desert (GD) and a previously validated chromosome 12 region were used as negative controls. The data was normalized to the chromosome 12 region and fold change was calculated relative to the average of WT clones. Data represents mean ± SD. *p* values were calculated by two-way ANOVA with Sidak correction for multiple comparisons (*n* = 4); \**p* < 0.05, \*\**p* < 0.01, \*\*\**p* < 0.001. *p* values for the comparisons are: CpIII: WT vs WT + IFNγ: *p* = 0.021, WT vs KO: *p* = 3.680E-04, WT + IFNγ vs KO + IFNγ: *p* = 1.328E-09; MTFP1: WT vs WT + IFNγ: *p* = 0.371, WT vs KO: *p* = 0.001, WT + IFNγ vs KO + IFNγ: *p* = 1.193E-06. (**B**) FAIRE-seq tracks depicting open chromatin peaks at CIITA pIII in two independent IFNγ-induced U2OS WT cells but not in the KO cells. The ZBTB48 ChIP-seq tracks are the same as described in Fig. 1A. (**C**) Schematic illustration for the proposed role of ZBTB48 as a priming factor at CIITA pIII. The model depicts the establishment of a nucleosome-depleted region at CIITA pIII upon binding of ZBTB48 binding at the ARE sites providing access to other transcription factors (TF) and chromatin remodeling complexes (CRC) upon IFNγ stimulation for transcription by pol II. Note that the two ARE-binding sites would roughly be placed side-by-side in a nucleosome-bound state. The model was created with BioRender. Source data are available online for this figure.

# Methods

## Reagents and tools table

| Reagent/resource | Reference or source | Identifier or catalog number |
| --- | --- | --- |
| **Experimental models** | | |
| U2OS cells (*H. sapiens*) | https://www.cellosaurus.org/CVCL_0042 | CVCL_0042 |
| HeLa Kyoto cells (*H. sapiens*) | https://www.cellosaurus.org/CVCL_1922 | CVCL_1922 |
| SUDHL4 cells (*H. sapiens*) | https://www.cellosaurus.org/CVCL_0539 | CVCL_0539 |
| HEK293T cells (*H. sapiens*) | https://www.cellosaurus.org/CVCL_0063 | CVCL_0063 |
| C57BL/6NTac (*M. musculus*) | Invivos | N/A |
| CD1 mice (*M. musculus*) | Charles River Laboratories | N/A |
| C57BL/6NTac ZBTB48 KO mice (*M. musculus*) | This study | N/A |
| **Recombinant DNA** | | |
| pCDNA3.1-FLAG-ZBTB48 | Jahn et al (2017) | N/A |

| Reagent/resource | Reference or source | Identifier or catalog number |
| --- | --- | --- |
| lentiCRISPRv2-puro | Addgene | 98290 |
| **Antibodies** | | |
| Rabbit anti-ZBTB48 | Atlas Antibodies | HPA030417 |
| Mouse anti-FLAG (M2) | Sigma-Aldrich | F1804 |
| Mouse anti-Tubulin | MPI-CBG Antibody Facility | N/A |
| Mouse anti-CIITA | Santa Cruz Biotechnology | sc-13556 |
| Mouse anti-HLA-DMB | Santa Cruz Biotechnology | sc-393548 |
| Mouse anti-CD74 | Santa Cruz Biotechnology | sc-6262 |
| Rabbit anti-H3K4me3 | Cell Signaling | 9751 |
| Rabbit IgG | ChromPure | 011-000-003 |
| Rabbit IgG | Bio-Rad | 1706515 |
| Mouse IgG | Bio-Rad | 1706516 |
| B220 (CD45R) [FITC] | eBioscience | RA3-6B2 |
| B220 (CD45R) [APC-Cy7] | Biolegend | RA3-6B2 |
| B220 (CD45R) [PE-Cy7] | BD | RA3-6B2 |
| CD3 [FITC] | BD | 145-2C11 |
| CD3e [APC-Cy] | Biolegend | 145-2C11 |
| CD34 [FITC] | eBioscience | RAM34 |

| Reagent/resource | Reference or source | Identifier or catalog number |
|---|---|---|
| CD4 [FITC] | BD | H129.19 |
| CD4 [APC-Cy7] | Biolegend | H129.19 |
| CD8a [APC-Cy7] | Biolegend | 53-6.7 |
| CD19 [APC-Cy7] | BD | 1D3 |
| CD19 [PE] | BD | 1D3 |
| CD24 [PE] | PE | M1/69 |
| CD41 [APC-Cy7] | Biolegend | MWReg30 |
| CD43 [FITC] | BD | S7 |
| CD71 [FITC] | BD | C2 |
| c-KIT [APC-Cy7] | eBioscience | 2B8 |
| c-KIT [PE-Cy7] | BD | 2B8 |
| FcgR (CD16/32) [BV510] | Biolegend | 2.4G2 |
| FcgR (CD16/32) [APC-Cy7] | BD | 2.4G2 |
| Gr1 [FITC] | eBioscience | RB6-8C5 |
| Gr1 [APC-Cy7] | Biolegend | RB6-8C5 |
| I-A/I-E [PE-Dazzle594] | Biolegend | M5 |
| IgD [PE] | BD | 11-26 c.2a |
| IgM [APC] | Southern Biotech | N/A |
| ILI7Ra (CD127) [FITC] | BD | A7R34 |
| Ly51 [Biotin] | BD | 6C3 |
| Mac-1 (CD11b) [APC] | Biolegend | M1/70 |
| Mac-1 (CD11b) [APC-Cy7] | eBioscience | M1/70 |
| Sca-1 [FITC] | BD | D7 |
| Sca-1 [APC] | eBioscience | D7 |
| Streptavidin [Pacific Blue] | eBioscience | N/A |
| Streptavidin [APC-Cy7] | BD | N/A |
| Ter119 [APC] | eBioscience | Ter119 |
| More information | This study | Dataset EV6 |
| **Oligonucleotides and other sequence-based reagents** | | |
| Primers for site-directed mutagenesis | This study | Dataset EV5 |
| Oligonucleotides for DNA pulldowns | This study | Dataset EV5 |
| Primers for RT-qPCR | This study | Dataset EV5 |
| Primers for ChIP-/FAIRE-qPCR | This study | Dataset EV5 |
| sgRNAs | This study | Dataset EV5 |
| Primers for T7EI assay/genotyping PCR | This study | Dataset EV5 |
| **Chemicals, enzymes, and other reagents** | | |
| DMEM, high glucose, pyruvate | Thermo Scientific | 11995073 |
| RPMI-1640 Medium | Thermo Scientific | 11875119 |
| Fetal Bovine Serum | Thermo Scientific | 10500064 |
| Opti-MEM I Reduced Serum Medium | Thermo Scientific | 31985070 |
| Polyethylenimine, Linear, MW 25000 | Polysciences | 23966 |
| Puromycin Dihydrochloride | Thermo Scientific | A1113802 |
| Hexadimethrine bromide, Polybrene | Sigma-Aldrich | H9268 |
| T7 Endonuclease I | New England BioLabs | M0302L |
| Q5 Hot Start High-Fidelity DNA Polymerase | New England BioLabs | M0493L |
| cOmplete, EDTA-free Protease Inhibitor Cocktail | Sigma-Aldrich | 11836170001 |
| T4 Polynucleotide Kinase | Thermo Scientific | EK0031 |
| T4 DNA Ligase | Thermo Scientific | EL0011 |
| Klenow Fragment, exo– | Thermo Scientific | EP0422 |
| ATP Solution | Thermo Scientific | R0441 |
| Biotin-7-dATP | Jena Bioscience | NU-835-BIO-S |

| Reagent/resource | Reference or source | Identifier or catalog number |
|---|---|---|
| illustra Microspin G-50 Columns | GE Healthcare | 27533001 |
| Sheared Salmon Sperm DNA | Life Technologies | AM9680 |
| IGEPAL CA-630 | Sigma-Aldrich | I8896 |
| Sample Buffer, Laemmli 2× Concentrate | Sigma-Aldrich | S3401 |
| 4x LDS sample buffer | Thermo Scientific | NP0007 |
| DTT | Sigma-Aldrich | 43816 |
| Iodoacetamide | Sigma-Aldrich | A3221-10VL |
| Sequencing-grade modified trypsin | Promega | V5113 |
| Reprosil-Pur C18-AQ 1.9 μm | Dr Maisch | r119.aq.0001 |
| Formaldehyde | Pierce | 28908 |
| Glycogen, MB Grade | Sigma-Aldrich | 1090139001 |
| Chloroform | Sigma-Aldrich | 366919-1L |
| UltraPurePhenol:Chloroform:Isoamyl Alcohol (25:24:1, v/v) | Life Technologies | 15593031 |
| IFNγ human | Sigma-Aldrich | I17001 |
| MycoAlert PLUS mycoplasma detection kit | Lonza | LT07-318 |
| 4–12% NuPAGE Novex Bis-Tris precast gel | Thermo Scientific | NP0321BOX |
| 12% NuPAGE Novex Bis-Tris precast gel | Thermo Scientific | NP0341BOX |
| SuperScript IV first-strand synthesis kit | Thermo Scientific | 18090200 |
| QuantiNova SYBR green PCR mix | Qiagen | 208056 |
| PVDF membrane | Bio-Rad | 1620177 |
| BCA Protein Assay Kit | Thermo Scientific | 23225 |
| Pierce ECL Western blotting substrate | Thermo Scientific | 32106 |
| Amersham ECL Select Western blotting reagent | GE Healthcare | RPN2235 |
| PageRule Plus Prestained Protein Ladder | Thermo Scientific | 26620 |
| MagicMarck XP Western Protein Standard | Thermo Scientific | LC5602 |
| MyTaq Red Polymerase | Bioline | BIO-21110 |
| Colloidal Blue Staining Kit | Thermo Scientific | LC6025 |
| TOPO TA cloning kit | Thermo Scientific | K250020 |
| Q5 Site-Directed Mutagenesis Kit | New England BioLabs | E0554S |
| Dynabeads MyOne Streptavidin C1 | Thermo Scientific | 65002 |
| Dynabeads Protein G magnetic beads | Thermo Scientific | 10004D |
| Qubit dsDNA HS Assay kit | Thermo Scientific | Q32856 |
| QIAamp DNA Blood Mini Kit | Qiagen | 51106 |
| QIAquick PCR Purification Kit | Qiagen | 28106 |
| Plasmid Midi Kit | Qiagen | 12143 |
| RNeasy Plus Mini Kit | Qiagen | 74134 |
| **Software** | | |
| Prism 8 | GraphPad | https://www.graphpad.com |
| MaxQuant [version 1.5.2.8] | Cox and Mann (2008) | https://www.maxquant.org |
| **Other** | | |
| SimpliAmp Thermal Cycler | Applied Biosystems | A24811 |
| QuantStudio 3/5 | Applied Biosystems | A28137 |
| ChemiDoc imaging system | Bio-Rad | N/A |
| FACS Aria II | BD | N/A |
| LSRII Flow Cytometer | BD | N/A |
| Concentrator Plus | Eppendorf | 5305000568 |

| Reagent/resource | Reference or source | Identifier or catalog number |
|---|---|---|
| Q Exactive HF | Thermo | N/A |
| EASY-nLC 1200 | Thermo | N/A |
| EpiShear Probe Sonicator | Active Motif | 53051 |

## Cell culture

U2OS (CVCL_0042), HeLa Kyoto (CVCL_1922), and HEK293T (CVCL_0063) were cultured in Dulbecco's Modified Eagle's Medium (DMEM) containing 4.5 g/l glucose, 4 mM glutamine, 1 mM sodium pyruvate and supplemented with 10% fetal bovine serum (FBS; Gibco), 100 U/ml penicillin and 100 μg/ml streptomycin (Gibco). SUDHL4 DLBCL cells (CVCL_0539) were cultivated in RPMI-1640 (Gibco) medium supplemented with 10% FBS, 2 mM glutamine, 100 U/ml penicillin and 100 μg/ml streptomycin. All three cell lines were maintained in a humidified incubator at 37 °C and 5% $CO_2$. For induction of CIITA, 250 U/ml IFNγ was added to the culture medium, and cells were incubated for 24 and 48 h for RNA and protein extraction, respectively. Cell lines were authenticated by STR profiling (1st BASE) and routinely monitored for mycoplasma contaminations using the MycoAlert PLUS mycoplasma detection kit (Lonza).

## Deletion mutant/variant construction

ZBTB48 zinc finger (ZnF) point mutants and deletion variants had been established previously (Jahn et al, 2017). pcDNA3.1-FLAG-ZBTB48 WT was used as a template for the generation of the constructs with point mutations in the C-terminal arm using the primers in Dataset EV5 and the Q5 Site-Directed Mutagenesis Kit (NEB). In brief, exponential amplification was performed with Q5 Hot Start High-Fidelity Master Mix according to the manufacturer's protocol using 10 ng of template with the following cycling conditions on a SimpliAmp thermal cycler (Applied Biosystems): 98 °C for 30 s, followed by 25 cycles with 98 °C for 10 s, 67 °C for 30 s, 72 °C for 3 min and a final extension at 72 °C for 3 min. The PCR product was phosphorylated and ligated with KLD enzyme mix (NEB) before transformation into competent cells followed by sequence verification (1st BASE) of the constructs.

## Transfection

Plasmids were transfected in U2OS cells using linear polyethylenimine (PEI, MW 25,000; Polysciences). 15.2 μg of plasmid and 60.8 μg of PEI were diluted in 2 ml of Opti-MEM, mixed, and incubated at room temperature for 20 min. The transfection mix was incubated with $4.5 \times 10^6$ cells seeded in a 15 cm dish for 7 h before washing with 1× PBS and replacement with fresh culture medium. The cells were collected 48 h post transfection for nuclear protein extraction.

For lentiviral packaging in HEK293T cells, each lentiviral transfer vector (0.5 μg) was co-transfected with 0.25 μg each of pMDLg/pRRE and pRSV-Rev and envelope vector pMD2.G (0.25 μg) into $3 \times 10^5$ cells in a 6-well plate. The plasmids were combined with 2.5 μg PEI in 100 μl Opti-MEM and incubated at room temperature for 20 min before incubating with cells for 18 h. The cells were washed with 1× PBS and replenished with fresh

culture medium. Lentiviral supernatants were collected 48 h after transfection and filtered with 0.22-μm syringe filters, and stored at −80 °C until use for transduction.

## Nuclear protein extraction

Nuclear extracts were prepared by incubating cells in hypotonic buffer (10 mM HEPES, pH 7.9, 1.5 mM $MgCl_2$, 10 mM KCl) on ice for 10 min. Cells were transferred to a dounce homogenizer in hypotonic buffer supplemented with 0.1% Igepal CA-630 (Sigma) and 0.5 mM DTT and lysed by 40 strokes. Nuclei were washed once in 1× PBS and incubated in hypertonic buffer (420 mM NaCl, 20 mM HEPES, pH 7.9, 20% glycerol, 2 mM $MgCl_2$, 0.2 mM EDTA, 0.1% Igepal CA-630 (Sigma), 0.5 mM DTT) for 2 h at 4 °C on a rotating wheel. Nuclear extract was collected by centrifugation at maximum speed for 1 h at 4 °C. Protein amounts were quantified with the Pierce BCA Protein Assay Kit according to the manufacturer's instructions (Thermo Scientific).

## DNA pulldown

As described previously (Jahn et al, 2017), equal amounts (25 μg) of forward and reverse sequence oligos (Dataset EV5) corresponding to each of the probes were combined in annealing buffer (20 mM Tris-HCl, pH 7.5, 10 mM $MgCl_2$, 100 mM KCl), denatured at 80 °C, and annealed by cooling. 100 units T4 kinase (Thermo Scientific) was added and incubated for 2 h at 37 °C for phosphorylation, followed by overnight incubation with 20 units T4 ligase. The concatenated DNA strands were purified using phenol–chloroform extraction, biotinylated with desthiobiotin-dATP (Jena Bioscience) and 60 units DNA polymerase (Thermo Scientific) and finally purified using microspin G-50 columns (GE Healthcare). The probes generated were immobilized on 250 μg paramagnetic streptavidin beads (Dynabeads MyOne C1, Thermo Scientific) on a rotation wheel for 30 min at room temperature. Subsequently, baits were incubated with 75 μg nuclear extract in PBB buffer (150 mM NaCl, 50 mM Tris-HCl pH 7.5, 5 mM $MgCl_2$, 0.5% Igepal CA-630 (Sigma) and 1 mM DTT) while rotating for 2 h at 4 °C. As a competitor for DNA binding, 20 μg sheared salmon sperm DNA (Ambion) was added. After three washes with PBB buffer, bound proteins were eluted in 2× Laemmli buffer (Sigma-Aldrich), boiled for 5 min at 95 °C and separated on a 4–12% NuPAGE Novex Bis-Tris precast gel (Thermo Scientific).

## Quantitative RT-PCR

RNA was isolated using the RNeasy Plus Mini Kit (Qiagen) and 1 μg of RNA was used for cDNA synthesis using the SuperScript IV first-strand synthesis kit with oligo-dT following the manufacturer's protocol. Quantitative PCR (qPCR) was carried out using 1x QuantiNova SYBR green PCR mix (Qiagen) and 500 nM each of the forward primer and reverse primer (Dataset EV5) with the following cycling conditions on either QuantStudio 3 or 5 Real-Time PCR systems (Applied Biosystems): 95 °C for 2 min, followed by 40 cycles with 95 °C for 5 s, 60 °C for 30 s followed by a melting curve. TBP was used as a housekeeping gene for normalization. A Ct (cycle threshold) value of 35 was imputed for samples with no Ct value to calculate fold changes. All data were analyzed using Prism 8 (GraphPad Prism, La Jolla, CA).

## Western blot

Protein samples were denatured in 2x Laemmli buffer (Sigma-Aldrich) at 95 °C for 5 min and separated on a 4–12% Bis-Tris gel (NuPAGE, Thermo Scientific) for 60 min at 170 V. For detection of CIITA, cells were lysed with 1× RIPA buffer supplemented with 1× cOmplete proteinase inhibitor (Roche) for 30 min on ice, pelleted and denatured with 1× LDS sample buffer (Thermo Scientific) at 95 °C for 5 min in the presence of 100 mM DTT. Protein amounts were quantified with the Pierce BCA Protein Assay Kit (Thermo Scientific) and at least 50 μg of protein were used. Samples were transferred to a PVDF membrane (Bio-Rad) using a semi-dry transfer system for 90 min at 50 mA per blot with 1x Transfer Buffer (25 mM Tris, 192 mM glycine, 0.02% (v/v) SDS, 10% (v/v) methanol). The membrane was blocked in PBS containing 0.1% Tween-20 (PBST) and 5% (w/v) nonfat milk for 1 h at RT prior to incubation with primary antibody (Dataset EV6) overnight at 4 °C. After three washes in 5% milk PBST for 10 min each at RT, the membrane was incubated for 60 min at RT with secondary antibodies (Dataset EV6) in 5% milk PBST, followed by two washes in 5% milk PBST, one wash in PBST 10 min each at RT and finally rinsed with 1× PBS before detection. The membrane was revealed with ECL substrate (Pierce ECL Western blotting substrate (Thermo Scientific) or Amersham ECL Select Western blotting detection reagent (GE Healthcare) using either a ChemiDoc imaging system (Bio-Rad) or X-ray films. PageRuler Plus Prestained Protein Ladder (Thermo Scientific) and MagicMark XP Western Protein Standard (Thermo Scientific) were used as size markers.

## Lentiviral transduction for generation of knockout cells

U2OS ZBTB48 WT and KO clones used here have been previously described (Jahn et al, 2017). In SUDHL4, CRISPR/Cas9 mediated ZBTB48 deletions were performed using three different sgRNA sequences (Dataset EV5) targeting its second exon. The sgRNAs were cloned into lentiCRISPRv2-puro vector and confirmed by Sanger sequencing (1st BASE). The lentiCRISPRv2-puro vector containing the Cas9 expression cassette and the gRNA scaffold was a gift from Brett Stringer (Addgene plasmid #98290; http://n2t.net/addgene:98290; RRID:Addgene_98290). For transduction, 700 μl of lentiviral supernatant, generated using HEK293T cells and supplemented with 10 mM HEPES and 8 μg/mL polybrene (Sigma-Aldrich), was combined with 100 μl of cell suspension containing $5 \times 10^5$ SUDHL4 cells in 12-well plates and spin-inoculated by centrifugation at 2000 rpm for 1.5 h at 32 °C. The reaction mix was replaced 24 h post transduction with fresh culture medium. Selection with 500 ng/ml puromycin was performed 48 h after transduction, over 2 weeks before T7 endonuclease assay (T7E1) assay. Upon confirmation of successful cleavage by Cas9, cells were single-sorted using a BD FACS Aria II flow cytometer and expanded to screen for ZBTB48 KO clones by Western blot. Similarly, SUDHL4 WT cells were also single-sorted and used as WT clones. The genotype of clones with undetectable levels of ZBTB48 and its downstream target, MTFP1, was further confirmed by Sanger sequencing. PCR products, amplified using gDNA of each clone using the same amplification protocol as the T7E1 assay, were cloned into the pCR2.1-TOPO TA vector (TOPO TA cloning

kit, Thermo Scientific) following the manufacturer's instructions and ten colonies from each clone were sequenced (1st BASE).

## T7E1 assay

A T7 endonuclease I assay was performed to evaluate Cas9-induced mutations. Genomic DNA was extracted from puromycin-selected SUDHL4 cells and untreated WT cells using the QIAamp DNA Blood Mini Kit (Qiagen) following the manufacturer's instructions. Using 100 ng of gDNA, a 835 bp region around the cut locus was amplified using the Q5 Hot Start High-Fidelity DNA Polymerase (NEB) and 400 nM of each primer (Dataset EV5) with the following PCR conditions: 94 °C for 3 min followed by 35 cycles with 94 °C for 30 s, 60 °C for 30 s, 72 °C for 50 s and a final extension at 72 °C for 5 min. PCR products were analyzed by gel electrophoresis and purified using QIAquick PCR purification kit (Qiagen). 100 ng purified DNA each from WT and sgRNA treated samples were denatured at 95 °C for 5 min in 1× NEB 2 buffer and hybridized following the temperature profile 95 to 85 °C at 2 °C/s and 85 to 25 °C at 0.1 °C/s. For digestion of heteroduplexes, 2 μl of 10× NEB 2 buffer, 1 μl T7 Endonuclease I (NEB) and 7 μl HPLC-grade $H_2O$ were added, and samples were incubated at 37 °C for 20 min. Cas9 activity was confirmed by separation of products on a 2% agarose gel.

## Generation of knockout mice

C57BL/6NTac and CD1 mice were obtained from Invivos, Singapore, and Charles River Laboratories, USA, respectively. All animal protocols were approved by the Institutional Animal Care and Use Committee (IUCAC) at the National University of Singapore, Singapore (protocol number: BR20-0926 and R20-0927). Animals were housed with food and water ad libitum, on a 12 h light/dark cycle (lights off at 7 pm) and the health status of the animals was routinely checked by a veterinarian.

Two sgRNAs (Dataset EV5) targeting regions at the opposite ends of exon 2 were designed using CRISPick (Broad Institute). The sgRNAs (Integrated DNA technologies, Singapore (IDT)) were tested in NIH3T3 mouse cell line prior to their use in the mice. CRISPR/Cas9 mediated transgenic mice were generated with the help from the Transgenic and Gene Targeting Facility at CSI, NUS. Briefly, equimolar amounts of crRNA and tracrRNA were annealed in microinjection buffer (MIB; 1 mM Tris, 0.25 mM EDTA pH 7.4) following the IDT protocol. The RNP (ribonucleoprotein) complex was prepared by incubating the reaction mix with equimolar amount of Alt-R S.p. Cas9 nuclease (# 1074181, IDT, Singapore) for 5 min at room temperature in MIB and then combined with ssODN to prepare the microinjection cocktail. Single-cell staged C57BL/6NTac mouse embryos were harvested from superovulated 4 week-old female mice at 0.5 d postcoitum after mating with male mice of the same strain (Brownstein, 2003). The microinjection cocktail was injected into either one of the pronuclei of C57BL/6NTac mouse embryos with Leica AM6000 manipulator using standard protocol (Brownstein, 2003). The zygotes were cultured overnight in KSOM mouse embryo culture medium and twenty-five viable zygotes were transferred to oviducts of 0.5 d postcoitum psedo-pregnant female CD1 mice. The pup tail clippings were genotyped by PCR following the protocol below. All transgenic mouse lines were back-crossed into C57BL/6NTac background and experiments

were performed on 8–14 week-old mice using littermate trios of −/−, +/−, +/+ of same sex.

## Mouse genotyping

In all, 2–3-week-old mice were ear tagged and tail clippings were collected for genotyping. For gDNA isolation, each tail tip was incubated in 500 µl of Lysis buffer (200 mM NaCl, 100 mM Tris-HCl pH 8.5, 5 mM EDTA, 0.2% SDS) with 2.5 µl of Proteinase K (20 mg/ml) overnight at 55 °C under constant agitation (800 rpm). 500 µl of digestion solution was added to the digestion solution and the precipitated DNA was fished out with a pipette tip and dissolved in 1x TE buffer for 2 h at 55 °C. Using 400 nM each of primers that flank the cut sites (Dataset EV5), PCR was performed using 1 µl of dissolved DNA, 1× MyTaq Red Reaction buffer red and 0.5 µl of MyTaq DNA polymerase (Bioline) on SimpliAmp thermal cycler (Applied Biosystems) using the following cycling conditions: 94 °C for 3 min followed by 35 cycles with 94 °C for 30 s, 60 °C for 30 s, 72 °C for 50 s and a final extension at 72 °C for 5 min. The genotype was determined based on the product size of the product by separation on a 1% agarose gel for 60 min at 100 V.

## Isolation and sorting of mouse cells

Retro-orbital bleeding was performed to collect 20 µl of blood from 8 to 12-week-old mice. The mice were euthanised, and spleen, thymus, long bones (tibias and femurs) and spine were collected for analysis. Viable single-cell suspension from the spleen and thymus was obtained by rubbing small pieces of the organ through a nylon filter (40 µm). BM cells were flushed from the long bones and spine with 1× PBS and next filtered through a nylon filter (35 µm) to obtain a single-cell suspension. The red blood cells from the cell suspension were lysed with Ammonium Chloride Potassium (ACK) red cell lysis buffer (150 mM Ammonium chloride, 10 mM Potassium bicarbonate, 0.1 mM EDTA) for 5 min at 4 °C. The lysis reaction was stopped by diluting the cell suspension mix to ten times the volume with 1× PBS, and the cells were pellet by centrifugation at 1000 rpm for 5 min and subsequently resuspended in 1× PBS for sorting and analysis. The RBC lysis step was performed twice for blood samples.

## Fluorescence-activated cell sorting (FACS) analysis

Hematopoietic stem and progenitor cells, and lineage-positive cells from BM, spleen and thymus of mice from each genotype were profiled by FACS. Littermates with at least one each of WT, Het and KO were analyzed together. Prior to antibody staining, the cells were blocked with FcgR (CD16/32) for 15 min on ice and then incubated with antibodies (Dataset EV6) for an additional 20 min on ice. 0.25 ng/mL of Hoechst 33258 was used to exclude dead cells. Samples were analyzed on a BD LSRII Flow Cytometer or a BD FACS Aria cell sorter.

## Protein expression and crystallization

The ZBTB48 ZnF10-11-C construct containing ZnF10-11 and the conserved C-terminal region (residues 548-620) was amplified from human brain cDNA library and cloned into a modified pGEX-4T-1 vector containing TEV (Tobacco Etch Virus) cleavage site. The proteins were induced in Escherichia coli BL21 (Gold) cells with 0.1 mM isopropyl β-D-1-thiogalactopyranoside (IPTG) for 5 h and purified by glutathione sepharose (GE Healthcare) followed by on-column TEV cleavage and finally by size-exclusion chromatography on Hiload 16/60 Superdex 75 coloumn (GE Healthcare).

The purified ZnF10-11-C protein was combined with double-stranded CIITA e.2 probe sequence (G-strand: 5′-CACAAGT-GAGGGATCA-3′; C-strand: 5′-GTGATCCCTCACTTGT-3′) at a 1:1.5 molar ratio in buffer containing 20 mM Tris-HCl (pH 7.5) and 500 mM NaCl. The reaction mix was dialyzed against a buffer containing 20 mM Tris-HCl (pH 7.5) and 150 mM NaCl and concentrated to 1 mM. Using the hanging drop vapor diffusion method, the crystals were grown at 20 °C by combining equal volumes of the protein-DNA complex and reservoir buffer (0.2 M imidazole malate, pH 8.5 and 20% (w/ v) polyethylene glycol 4 K). Reservoir buffer with 25% glycerol was used as cryoprotectant.

X-ray diffraction data collection was performed on beamline 19U1 (BL19U1) at the Shanghai Synchrotron Radiation Facility (SSRF) at wavelength 0.9758 and *HKL2000* software (HKL Research) (Otwinowski and Minor, 1997). The initial crystallographic phases were calculated using molecular replacement which was carried out by Phaser employing the ZBTB48-telomeric DNA complex structure as the search model (PDB code: 5YJ3) (Adams et al, 2010; McCoy et al, 2007) and the model was further built and refined using Coot and Phenix.refine (Adams et al, 2010; Emsley et al, 2010).

## Next-generation RNA sequencing and data analysis

RNA was extracted from IFNγ (250U/ml, 24 h) treated and untreated control U2OS WT and ZBTB48 KO clones using the RNeasy Plus Mini Kit (Qiagen) with additional on-column DNaseI digestion following the manufacturer's instructions. Five clones per condition were used as biological replicates. RNA quality assessment, library preparation and sequencing was performed at NovogeneAIT (Singapore). RNA-seq (stranded mRNA) library prepared after quality assessment (RIN# > 7) were sequenced on a NovaSeq 6000 PE150 and 53 to 82 million reads were obtained per sample. Reads were aligned to the human reference genome version GRCh38 (GCA_000001405.15) using STAR v2.7.9a (Dobin et al, 2013) whose index was generated with RefSeq annotation and up to 4 mismatches were allowed per pair. The alignments were filtered for unique mappers using NH:i:Nmap field with value 1 and further quantified using featureCounts v2.4.0 (Liao et al, 2014). DESeq2 was used for differential expression analysis and log fold changes were shrunk using the original DESeq2 shrinkage estimator (normal) (Love et al, 2014). Bigwig tracks were generated using bamCoverage (deeptools v3.5.1) (Ramírez et al, 2016), hosted on cyverse (Merchant et al, 2016) and visualized on UCSC Genome Browser (Kent et al, 2002).

## MS sample preparation

Protein samples were boiled in Laemmli buffer (Sigma-Aldrich) at 95 °C for 5 min and separated on a 12% Bis-Tris gel (NuPAGE, Thermo Scientific) for 30 min at 170 V. Proteins were stained using the Colloidal Blue Staining Kit (Thermo Scientific) according to the manufacturer's instructions. For in-gel digestion, samples were separated in 4 equal gel pieces for each sample from high to low molecular weight, and each fraction was cut individually with a clean scalpel into 1 mm × 1 mm pieces. First, gel pieces were

destained in destaining buffer (50% 50 mM ammonium bicarbonate buffer, 50% ethanol) twice, followed by two rounds of dehydration in 100% acetonitrile. Supernatants were discarded and residual liquid removed using a Concentrator Plus (Eppendorf). Samples were then reduced in 10 mM DTT (Sigma) for 1 h at 56 °C followed by alkylation with 55 mM iodoacetamide (Sigma) for 45 min in the dark. Tryptic digest was performed in 50 mM ammonium bicarbonate buffer with 2 µg sequencing-grade trypsin (Promega) at 37 °C overnight. Supernatants were collected and the digested peptides extracted from the gel pieces by one round of extraction buffer (30% acetonitrile, 10% trifluoracetic acid, 70% mM ammonium bicarbonate buffer), one round of 100% acetonitrile, another round of extraction and two more rounds of 100% acetonitrile for 15–20 min at RT with shaking. All supernatants were combined and evaporated in a Concentrator Plus (Eppendorf) to reduce the volume to <200 µl. Peptides were desalted on self-made C18 Stage Tips and analyzed with a Q Exactive HF mass spectrometer (Thermo) coupled to an EASY-nLC 1200 system (Thermo) nanoflow liquid chromatography system. Peptides were separated on a C18 reversed-phase capillary (25 cm long, 75-µm inner diameter, packed in-house with ReproSil-Pur C18-AQ 1.9-µm resin (Dr. Maisch) directly mounted on the electrospray ion source. We used a 215-min gradient from 2 to 40% acetonitrile in 0.5% formic acid at a flow of 225 nl/min. The Q Exactive HF was operated in positive ion mode with a Top20 MS/MS spectra acquisition method per MS full scan. MS scans were obtained with 60,000 resolution at a maximum injection time of 20 ms and MS/MS scans at 15,000 resolution with maximum IT of 75 ms.

## MS data acquisition and data analysis

The raw files were processed with MaxQuant (Cox and Mann, 2008) version 1.5.2.8 against the UNIPROT annotated human protein database (81,194 entries). Carbamidomethylation on cysteines was set as fixed modification while methionine oxidation and protein N-acetylation were considered as variable modifications. Peptide and protein FDR were enforced at 0.01 with match between runs option activated. MaxLFQ quantitation (Cox et al, 2014) was based on unique peptides for protein groups with at least 2 ratio counts. The resulting protein groups table was further processed with an in-house script to exclude contaminants (based on the MaxQuant contaminant list), reverse hits and proteins that were only identified by a modified peptide. Missing values were imputed based on normally distributed values corresponding to the percentage of missing values in each sample for proteins that were quantified in at least four out of five biological replicates.

## Chromatin immunoprecipitation (ChIP) and quantitative PCR

Cells were washed twice with ice-cold 1× PBS and crosslinked for 20 min at RT with 1% (v/v) formaldehyde (Pierce) diluted in respective cell culture media. The reaction was quenched by the addition of Glycine-PBS (0.125 M final concentration) for 5 min at RT and washed with ice-cold 1× PBS. The fixed cells were collected by scraping in 0.5 ml of ice-cold 1× PBS supplemented with 1× cOmplete proteinase inhibitor (Roche) for U2OS and by centrifugation for SUDHL4 at $1000 \times g$ for

5 min at 4 °C. The pellets were snap-frozen and stored at in −80 °C until lysis. For lysis, cells were resuspended and incubated at 4 °C on a rotating wheel in Lysis buffer 1 (50 mM Tris-HCl pH 8.0, 250 mM sucrose, 140 mM NaCl, 1 mM EDTA pH 8.0, 10% glycerol, 0.5% Igepal CA-630, 0.25% Triton X-100, 0.25% Tween-20, 1× cOmplete proteinase inhibitor) for 15 min, followed by Lysis buffer 2 (10 mM Tris-HCl pH 8.0, 200 mM NaCl, 1 mM EDTA pH 8.0, 0.5 mM EGTA pH 8.0, 1× cOmplete proteinase inhibitor) for 10 min with centrifugation at $1000 \times g$ for 10 min at 4 °C in between. The nuclear pellet was collected by centrifugation and resuspended in sonication buffer (1% SDS, 10 mM EDTA pH 8.0, 50 mM Tris-HCl pH 8.0, 1× cOmplete proteinase inhibitor). The chromatin in sonication buffer was sheared to 200–400 bp by sonication using an EpiShear Probe Sonicator (Active Motif) at 30% amplitude, 25 cycles of 15 s ON/30 s OFF for U2OS and 30 cycles for SUDHL4. Lysates containing 25 µg of sheared chromatin were incubated with 0.833 µg of antibodies (Dataset EV6) in PBB1 buffer (180 mM NaCl, 50 mM Tris-HCl pH 7.5, 5 mM MgCl$_2$, 1 mM DTT, 0.5% IGEPAL CA-630, 1× cOmplete proteinase inhibitor) overnight at 4 °C on a rotating wheel. Prior to use, for each ChIP reaction 12.5 µl of Dynabeads protein G magnetic beads (Thermo Scientific) were blocked with 33.3 µg of sheared salmon sperm DNA (Thermo Scientific) in PBB1 containing 10 µg/µl BSA for 10 min at room temperature on rotating wheel. The pre-blocked beads were mixed with the ChIP reactions containing antibodies and incubated for 2 h at 4 °C on a rotating wheel followed by seven washes with PBB2 buffer (180 mM NaCl, 50 mM Tris-HCl pH 7.5, 5 mM MgCl$_2$, 1 mM DTT, 0.5% IGEPAL CA-630, 1× cOmplete proteinase inhibitor) and once with ice-cold 1× TE buffer. DNA was eluted twice in 100 µl of elution buffer (0.1 M NaHCO$_3$, 1% SDS) and de-crosslinked overnight at 65 °C in the presence of 0.2 M NaCl. The DNA samples were treated with 0.5 µg/µl of RNase A for 30 min at 37 °C and then with 0.25 µg/µl of Proteinase K (Roche) in the presence of 10 mM EDTA, 40 mM Tris-HCl pH 6.5 for 2 h at 45 °C prior to purification with QIAquick PCR Purification Kit (Qiagen). For "input" DNA, 2.5 µg (10% of ChIP reaction) of sheared chromatin lystes were treated similarly to elutes. qPCR was performed using ChIP primers indicated in Dataset EV5 with the same amplification protocol mentioned in "qPCR" section.

Next-generation ChIP-sequencing (ChIP-seq) was performed by NovogeneAIT (Singapore). The purified DNA samples were assessed for quality prior to library preparation and sequenced on a NovaSeq 6000 PE150 and 41 to 53 millions of 150 bp paired-end reads were obtained per sample. Reads were trimmed for adapter sequences using trimmomatic PE v0.39 (Bolger et al, 2014) and aligned to the human reference genome version GRCh38 using bowtie2 version 2.3.5.1 (Langmead and Salzberg, 2012) with default settings and processed (sorting and indexing) using samtools version 1.12 (Danecek et al, 2021). Multimapping and unmapped reads were filtered out and duplicate reads were removed using sambamba version 1.0 (Tarasov et al, 2015). Peaks were called using MACS version 2.1.1.2 (Zhang et al, 2008) in paired-end mode with default q-value cut-off (0.05) and --broad flag. We performed a quantitative differential binding analysis with Diffbind 3.2.1 (Ross-Innes et al, 2012) between the WT and ZBTB48 KO conditions. Peaks which were significantly different (FDR < 0.01 and fold change > 2) and called consistently in all 3 WT replicates were marked as "TRUE" in the "DB" column. The resulting peak set was annotated with the closest TSS using the hg38 annotation database generated from UCSC. Bigwig tracks normalized to counts per million mapped reads were generated using bamCoverage (deeptools v3.5.0)

(Ramírez et al, 2016), hosted on cyverse (Merchant et al, 2016) and visualized on UCSC Genome Browser (Kent et al, 2002).

## Formaldehyde-assisted isolation of regulatory elements (FAIRE)

FAIRE was performed by phenol–chloroform extraction of the sonicated 1% formaldehyde fixed samples obtained as described in the ChIP section. For collection of "free chromatin" 30 µg of sonicated samples was treated with 10 µg/µl of RNase A (Thermo Scientific) in the presence of 0.2 M NaCl at 37 °C for 1.5 h. DNA was obtained after two extractions with phenol–chloroform-isoamyl alcohol followed by a wash with chloroform followed by precipitation in the presence of 0.5 M NaCl, 40 µg of glycogen and equal volume of isopropanol at −80 °C for at least 1 h. After final precipitation by 70% ethanol, the DNA pellet was resuspended in 30 µl of nuclease-free water and incubated at 65 °C overnight for de-crosslinking. Input DNA was obtained following the sample phenol–chloroform purification steps post overnight de-crosslinking of 10 µg of sonicated sample at 65 °C in the presence of 0.2 M NaCl and 10 µg/µl of RNase A and 0.8 µg/µl of Proteinase K (Roche). DNA was quantified with the Qubit dsDNA HS Assay Kit (Thermo Scientific) according to the manufacturer's instructions. qPCR was performed with 10 ng of DNA using the same primers and amplification protocol as for ChIP (Dataset EV5) and relative free DNA was estimated as outlined by Rodriguez-Gil et al (Rodríguez-Gil et al, 2018).

For FAIRE-seq, free chromatin and input DNA was assessed for quality prior to library preparation and sequencing at NovogeneAIT (Singapore). Sequencing was performed on a NovaSeq 6000 PE150 and 41 to 62 millions of 150 bp paired-end reads were obtained per sample. Reads were stripped of adapter sequences with trimmomatic PE v0.39 (Bolger et al, 2014) and mapped to human reference genome (GRCh38) using bowtie2 v2.2.5 (Langmead and Salzberg, 2012), while reads that had a mapping quality less than 30 were removed using samtools v1.14 (Danecek et al, 2021). Peaks were called using MACS 2.2.7.1 (Zhang et al, 2008) in paired-end mode and subject to irreproducible discovery rate analysis (IDR 2.0.4.2) (Li et al, 2011) wink rank based on $p$ value and allowing non-overlapping peaks to measure reproducibility between replicates. Further, reproducible regions (IDR score >125) in replicates of U2OS WT and ZBTB48 KO, respectively, were considered to determine areas of enrichment by calculating the RPKM and log2 fold change at these regions using read counts reported by multiBamCov (bedtools v2.30.0) (Quinlan and Hall, 2010). Bigwig tracks normalized to counts per million mapped reads were generated using bamCoverage (deeptools v3.5.1), hosted on cyverse (Merchant et al, 2016) and visualized on UCSC Genome Browser (Kent et al, 2002).

## Experimental study design and statistics

No sample-size calculation was performed. The number of independent biological replicates (stated in the figure legends) are based on on experience from similar experiments and consistent with the current practise in the field. Samples were not randomized in our analyses. We did not blind operators during the analysis. No data were excluded from the analysis.

## Data availability

The atomic coordinates and structure factors for the ZBTB48 DNA-binding domains in complex with its CIITA promoter binding site have been deposited to the Protein Data Bank (PDB) under the accession code 8JQ0 (please refer to the attached PDB Structure Validation Report; https://www.rcsb.org/structure/unreleased/8JQ0). The mass spectrometry data have been deposited to the ProteomeXchange Consortium via the PRIDE partner repository (Perez-Riverol et al, 2021) with the dataset identifier PXD043134 (https://www.ebi.ac.uk/pride/login; username: reviewer_pxd043134@ebi.ac.uk; password: Clu0kTCB). The RNA-seq, ChIP-seq and FAIRE-seq data are available at the NCBI Gene Expression Omnibus (GEO) (Barrett et al, 2013) with the identifier GSE237959 (reviewer token: ufwxsucszrsjnip). All other newly generated materials and reagents will be provided by the authors upon request.

The source data of this paper are collected in the following database record: biostudies:S-SCDT-10_1038-S44318-024-00306-y.

## Peer review information

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

## Acknowledgements

The authors are grateful to all members of the Kappei lab for advice and discussions. This research was supported by the National Research Foundation Singapore and the Singapore Ministry of Education under its Research Centres of Excellence initiative, funded by the Singapore Ministry of Education's Tier 3 grants (MOE2014-T3-1-006), funding by the Fritz Thyssen Foundation (10.21.1.022MN) and an NMRC Open Fund Individual Research Grant (MOH-OFIRG21jun-011). The authors thank the Fluorescence Activated Cell Sorting (FACS) Facility and the Quantitative Proteomics Core (QPC) at the Cancer Science Institute of Singapore (CSI) for their support.

## Author contributions

**Grishma Rane**: Conceptualization; Data curation; Formal analysis; Investigation; Visualization; Methodology; Writing—original draft; Writing—review and editing. **Vivian L S Kuan**: Data curation; Formal analysis; Investigation; Visualization; Methodology; Writing—review and editing. **Suman Wang**: Data curation; Formal analysis; Investigation; Methodology; Writing—review and editing. **Michelle Meng Huang Mok**: Data curation; Formal analysis; Investigation; Methodology; Writing—review and editing. **Vartika Khanchandani**: Data curation; Formal analysis; Investigation; Visualization; Methodology; Writing—review and editing. **Julia Hansen**: Data curation; Formal analysis; Investigation; Methodology; Writing—review and editing. **Ieva Norvaisaite**: Data curation; Formal analysis; Investigation; Methodology; Writing—review and editing. **Naasyidah Zulkaflee**: Data curation; Formal analysis; Investigation; Methodology; Writing—review and editing. **Wai Khang Yong**: Investigation; Methodology; Writing—review and editing. **Arne Jahn**: Resources; Methodology; Writing—review and editing. **Vineeth T Mukundan**: Data curation; Formal analysis; Writing—review and editing. **Yunyu Shi**: Resources; Supervision; Funding acquisition; Writing—review and editing. **Motomi Osato**: Data curation; Supervision; Writing—review and editing. **Fudong Li**: Data curation; Formal analysis; Supervision; Funding acquisition; Investigation; Methodology; Writing—review and editing. **Dennis Kappei**: Conceptualization; Data curation; Formal analysis; Supervision; Funding acquisition; Investigation; Visualization; Methodology; Writing—original draft; Project administration; Writing—review and editing.

Source data underlying figure panels in this paper may have individual authorship assigned. Where available, figure panel/source data authorship is listed in the following database record: biostudies:S-SCDT-10_1038-S44318-024-00306-y.

## Disclosure and competing interests statement

The authors declare no competing interests.

# Expanded View Figures

**Figure EV1.  ZBTB48 regulates IFNγ-induced CIITA pIII expression.**

(A) ChIP-seq tracks depicting ZBTB48 binding peaks at CIITA pIII in HeLa cells but not in the KO cells ($n = 4$). No reads are observed in the corresponding RNA-seq tracks in these uninduced cells ($n = 5$; data re-analyzed from Jahn et al (Jahn et al, 2017)). (B) Relative mRNA expression of total (pan-)CIITA and the three promoter-specific transcripts in each five independent HeLa WT and ZBTB48 KO clones treated with or without 250 U/ml IFNγ for 24 h. Data represents mean ± SEM. Data points with gray circles are imputed due to lack of detectable Ct values. The $\log_2$ fold change is calculated relative to the average of five untreated WT clones. *p* values were calculated by 2-way ANOVA with Sidak correction for multiple comparisons ($n = 5$); ***$p < 0.001$. pan-CIITA: WT vs WT + IFNγ: $p < 1E-15$, KO vs KO + IFNγ: $p < 1E-15$, WT vs KO: $p = 1.000$, WT + IFNγ vs KO + IFNγ: $p = 0.941$; CIITA pIII: WT vs WT + IFNγ: $p < 1E-15$, KO vs KO + IFNγ: $p = 5.386E-12$, WT vs KO: $p = 0.999$, WT + IFNγ vs KO + IFNγ: $p < 1E-15$; CIITA pIV: WT vs WT + IFNγ: $p = 2.260E-13$, KO vs KO + IFNγ: $p = 2.965E-11$, WT vs KO: $p = 0.712$, WT + IFNγ vs KO + IFNγ: $p = 0.077$. (C) Differential expression analysis of the RNA-seq quantification comparing 5 U2OS WT clones $+/-$ IFNγ treatment (250 U/ml IFNγ for 24 h). Genes belonging to the CIITA-MHC-II family are in blue and the rest of the IFNγ response genes (ISGs) in yellow. Additional proteins that expressed above a two-dimensional cut-off of >32-fold enrichment (owing to the large number of strongly induced genes) and $p < 0.01$ are in salmon. (D) Differential expression analysis of the RNA-seq quantification comparing 5 U2OS ZBTB48 KO clones $+/-$ IFNγ treatment (250 U/ml IFNγ for 24 h) depicted as in (C). (E) GO term analysis for enriched genes in pair-wise comparisons in Fig. 3D, Fig. EV1C,D, and Fig. EV1G depicting significantly enriched GO terms. The size of the dots corresponds to the number of genes representing each term. (F) Heatmap of differentially expressed MHC-II family genes detected in RNA-seq data comparing 5 U2OS WT and ZBTB48 KO clones $+/-$ IFNγ. (G) Differential expression analysis of the RNA-seq quantification comparing 5 U2OS WT and ZBTB48 KO clones (without IFNγ treatment). Genes are colored as in (C, D), but here a >twofold enrichment cut-off was applied (as in Fig. 3D) given that baseline expression differences are compared.

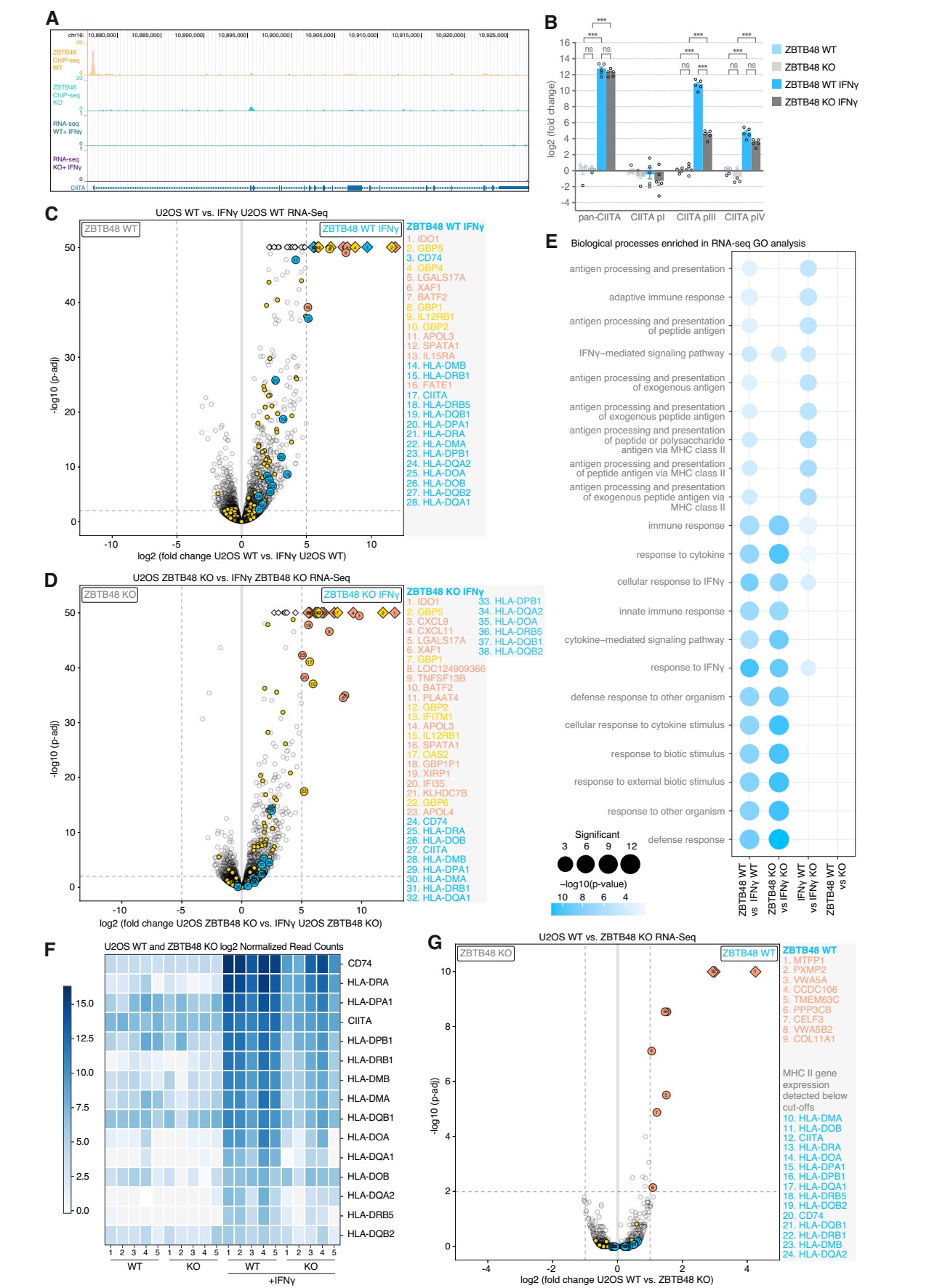

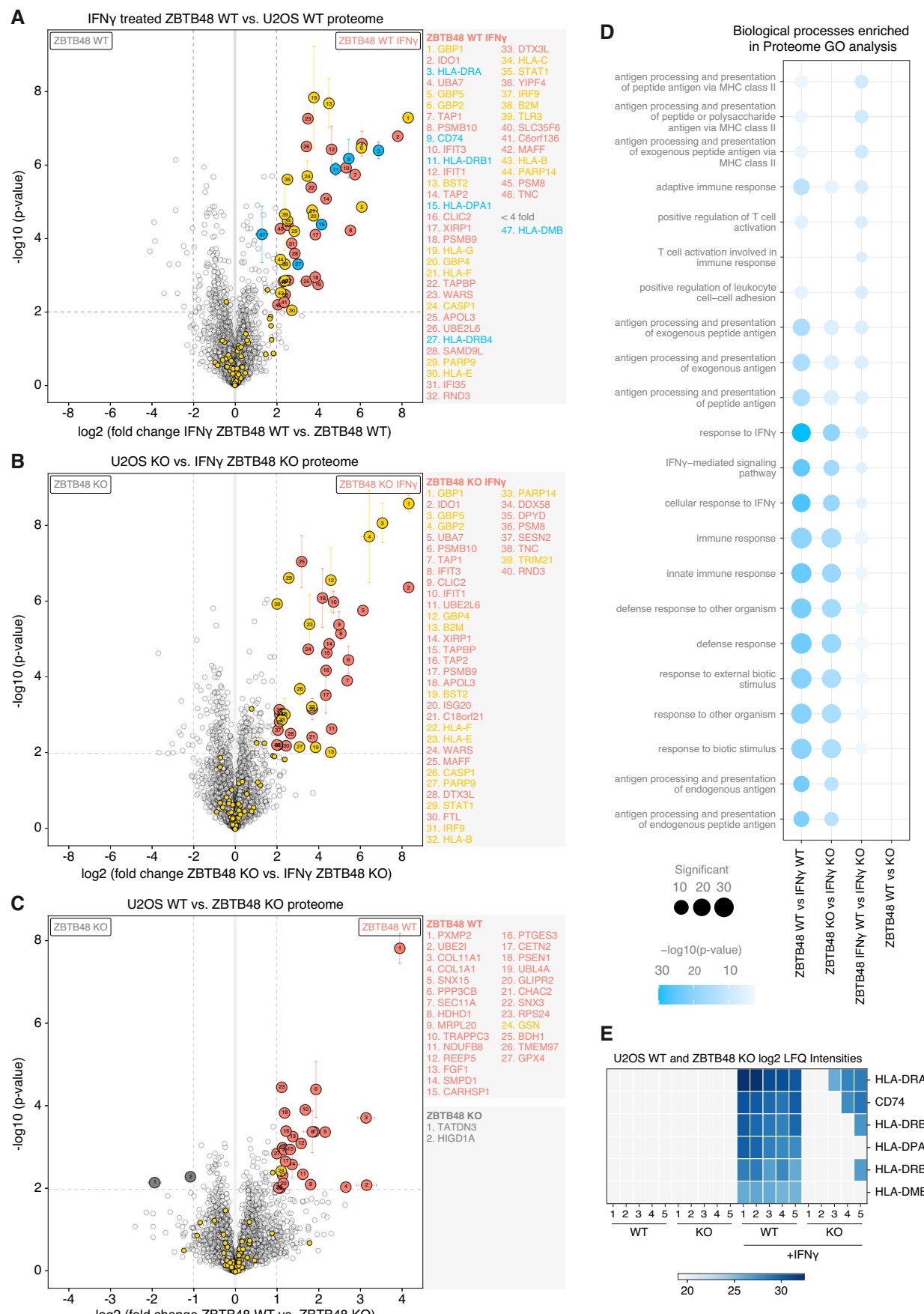

◀    **Figure EV2.   ZBTB48 exclusively affects the CIITA-MHC-II expression program upon IFNγ stimulation at the protein level.**

(**A**) Differential expression analysis of label-free quantitative protein expression comparing 5 U2OS WT clones +/− IFNγ treatment (250 U/ml IFNγ for 48 h). Genes belonging to the CIITA-MHC-II family are in blue and the rest of the IFNγ response genes (ISGs) in yellow. Additional proteins that expressed above a two-dimensinal cut-off of >4-fold enrichment (owing to the large number of strongly induced genes) and $p < 0.01$ are in salmon. (**B**) Differential expression analysis of label-free quantitative protein expression comparing 5 U2OS ZBTB48 KO clones +/− IFNγ treatment (250 U/ml IFNγ for 48 h) depicted as in (**A**). (**C**) Differential expression analysis of label-free quantitative protein expression comparing 5 U2OS WT and ZBTB48 KO clones (without IFNγ treatment). Genes are colored as in (**A, B**), but here a > 2-fold enrichment cut-off was applied (as in Fig. 3D) given that baseline expression differences are compared. (**A–C**) Two-dimensional error bars represent the standard deviation based on iterative imputation cycles during the label-free analysis to substitute missing values (e.g., no detection in the KO clones) and the measure of center is mean. $p$ values were calculated by independent sample $t$ test ($n = 5$). (**D**) GO term analysis for enriched proteins in pair-wise comparisons in Figs. 3F and  EV2A–C depicting significantly enriched GO terms. The size of the dots corresponds to the number of genes representing each term. (**E**) Heatmap of differentially expressed MHC-II family genes detected in the proteome data comparing 5 U2OS WT and ZBTB48 KO clones +/− IFNγ.

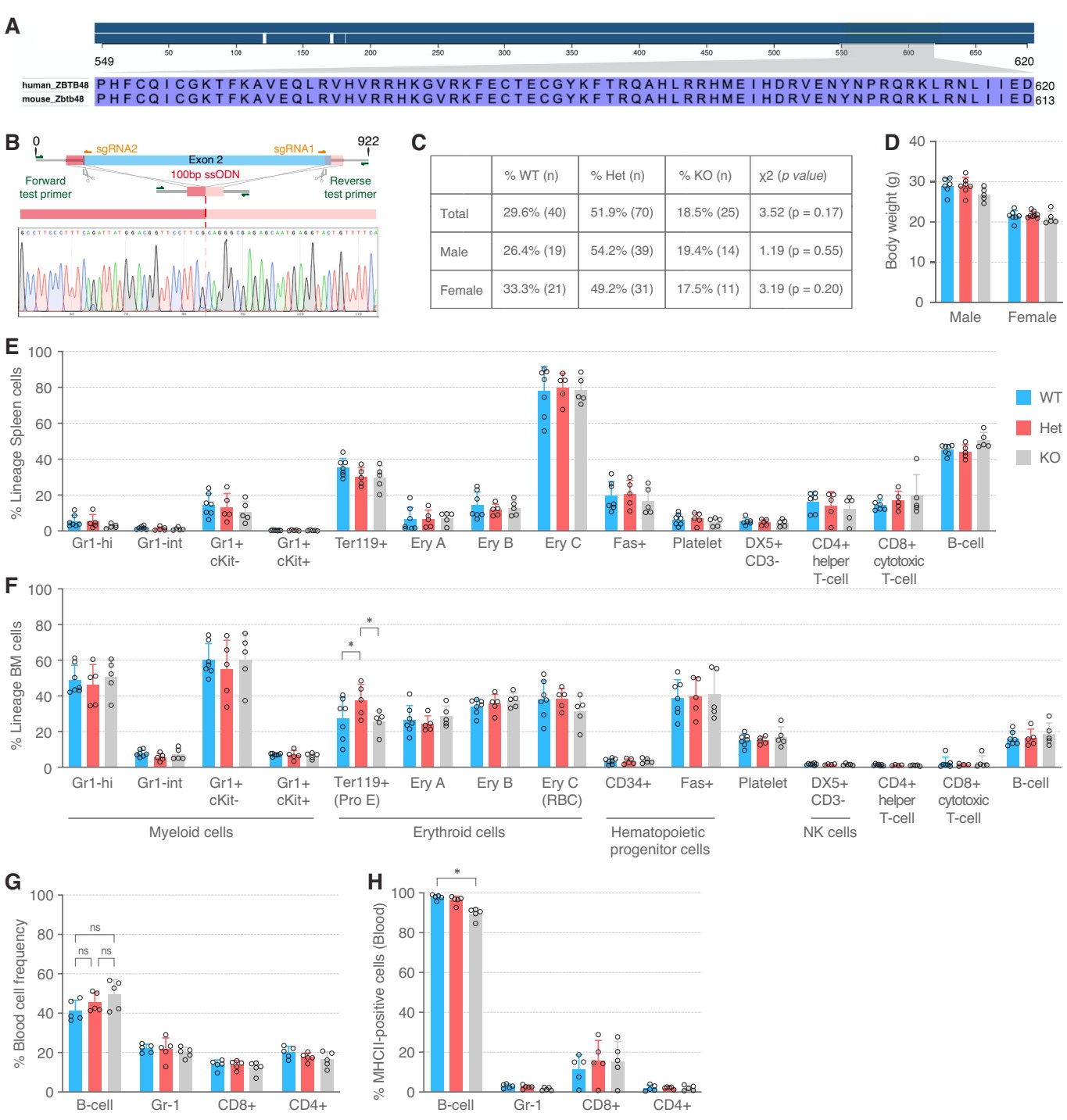

**Figure EV3.  Loss of ZBTB48 does not affect abundance of blood cell populations.**

(A) Protein sequence alignment using Clustal Omega of human and mouse ZBTB48 (top) with a zoom in to the DNA-binding region including ZnF10, ZnF11 and the C-terminal arm that are required for binding to CIITA pIII (bottom). (B) Schematic of the sgRNA and ssODN design used for creation of the ZBTB48 KO strain (top) and Sanger sequencing verification of successful and precise deletion (bottom) in ZBTB48$^{-/-}$ animals. (C) $\chi^2$-test comparing experimentally determined off-spring frequency in crosses between ZBTB48$^{+/-}$ animals. The genotype distribution follows Mendelian inheritance patterns for the total cohort (top), male (middle) and female (bottom) mice. (D) Body weight in g calculated separately for males and females across the three genotypes. Data represents mean ± SD (male: WT $n = 6$, Het $n = 7$, KO $n = 5$; female: WT $n = 7$, Het $n = 8$, KO $n = 5$). These data were used to generate the ratio with the spleen weight in Fig. 4D. (E) Percentage of spleen lineage cells in WT ($n = 7$), Het ($n = 5$) and KO ($n = 5$) mice. (F) Percentage of bone marrow lineage cells in WT ($n = 7$), Het ($n = 5$) and KO ($n = 5$) mice. (G) Percentage of blood cell frequencies in WT ($n = 7$), Het ($n = 5$) and KO ($n = 5$) mice. (H) Percentage of MHC-II-positive cells in the blood cell types from Fig. EV4F. For (E–H) data represents mean ± SD. $p$ values were calculated by two-way ANOVA with Sidak correction for multiple comparisons; *$p < 0.05$. $p$ values for the percentage cells in (F): WT vs Het: $p = 0.045$, Het vs KO: $p = 0.021$; in (H): WT vs KO: $p = 0.03$. Gr1-hi, Gr1 high; Gr1-int, Gr1 intermediate.

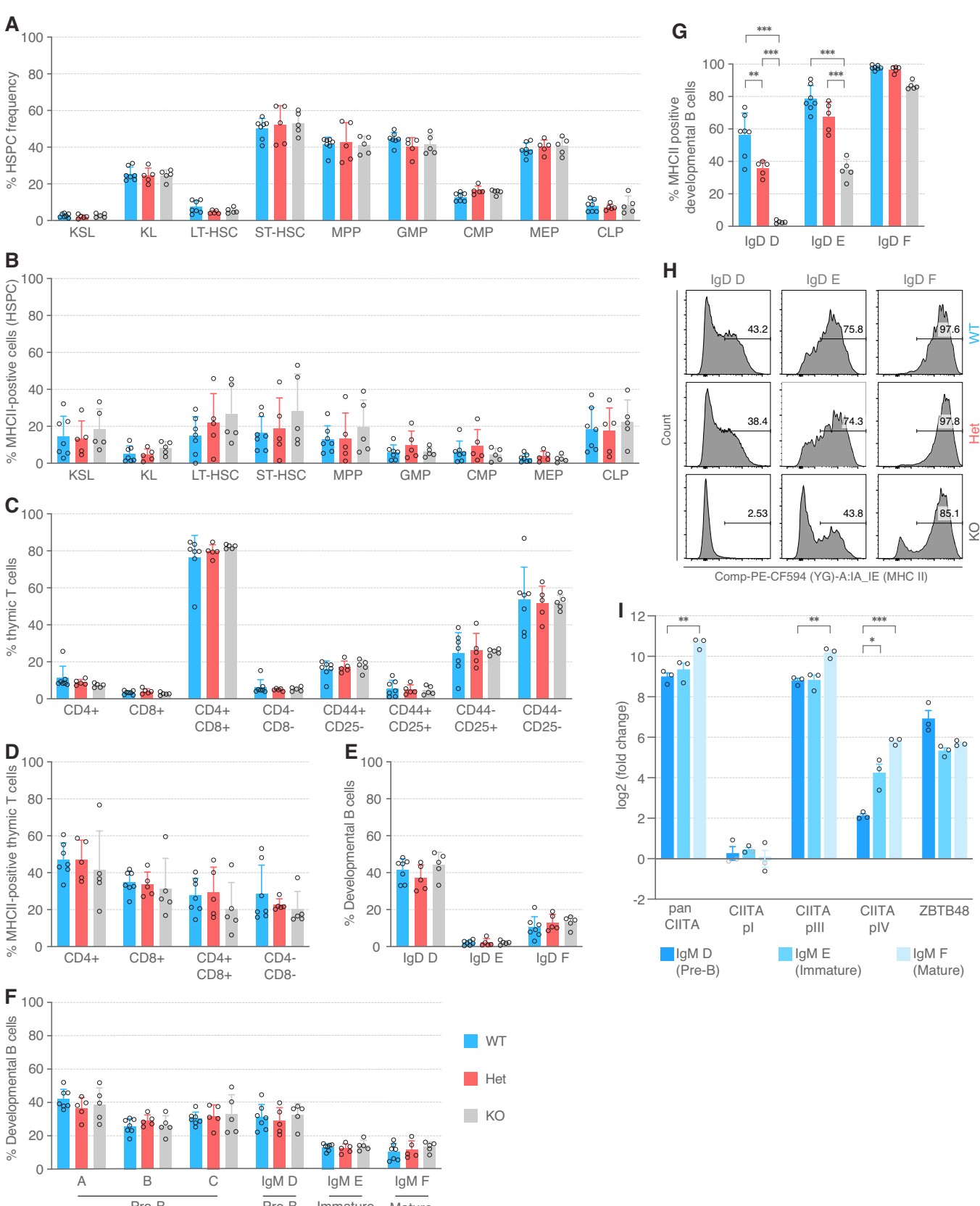

◀ **Figure EV4. Loss of ZBTB48 does not affect abundance of hematopoietic stem cells, developmental B cells and thymic T cells.**

(A) Percentage of hematopoietic stem and progenitor cell (HSPC) types in WT ($n = 7$), Het ($n = 5$) and KO ($n = 5$) mice. KSL: c-kit⁺ Sca-1⁺ Lineage⁻ (Lin⁻), KL: c-kit⁺ Lin⁻, LT-HSC: long-term 'quiescent' hematopoietic stem cells, ST-HSC: short-term 'cycling' hematopoietic stem cells, MPP: multipotent progenitor (KSL CD34⁺ flt3⁺), GMP: granulocyte-macrophage progenitor, CMP: common myeloid progenitor, MEP: megakaryocyte-erythroid progenitor, CLP: common lymphoid progenitor. (B) Percentage of MHC-II-positive cells in HSPCs from Fig. EV4A. (C) Percentage of thymic T cells in WT ($n = 7$), Het ($n = 5$) and KO ($n = 5$) mice. (D) Percentage of MHCII-positive cells in thymic T cells from Fig. EV4C. (E) Percentage of development B cell types in WT ($n = 7$), Het ($n = 5$) and KO ($n = 5$) mice when gated by IgD. (F) Percentage of development B cell types in WT ($n = 7$), Het ($n = 5$) and KO ($n = 5$) mice when gated by IgM. The developmental stage of the B cells is indicated below. (G) Percentage of MHC-II-positive developmental B cells in WT ($n = 7$), Het ($n = 5$) and KO ($n = 5$) mice when gated by IgD in (E). For (A–G) data represents mean ± SD. $p$ values were calculated by two-way ANOVA with Sidak correction for multiple comparisons; **$p < 0.01$, ***$p < 0.001$. $p$ values for the percentage MHC-II cells comparisons are as follows: D (pre-B): WT vs Het: $p = 0.001$, WT vs KO: $p = 3.300E-14$, Het vs KO: $p = 1.238E-06$; E (immature): WT vs KO: $p = 1.223E-11$, Het vs KO: $p = 1.598E-06$. (H) Representative flow cytometry analysis of MHC-II-positive cells in D (pre-B), E (immature) and F (mature) subpopulations from WT, Het and KO littermates when gated by IgD. (I) Relative mRNA expression of total (pan-)CIITA, the three promoter-specific transcripts and ZBTB48 mRNA in D (pre-B), E (immature), and F (mature) subpopulations from WT mice. Data represents mean ± SEM. Data points with gray circles are imputed due to lack of detectable Ct values. The $log_2$ fold change is calculated relative to samples with imputed values. $p$ values were calculated by multiple $t$ tests with Holm-Sidak correction for multiple comparisons ($n = 3$); *$p < 0.05$, **$p < 0.01$, ***$p < 0.001$. For D vs F: pan-CIITA: $p = 0.008$, CIITA pIII: $p = 0.007$, CIITA pIV: $p = 6.100E-05$; For D vs E: CIITA pIV: $p = 0.05$.

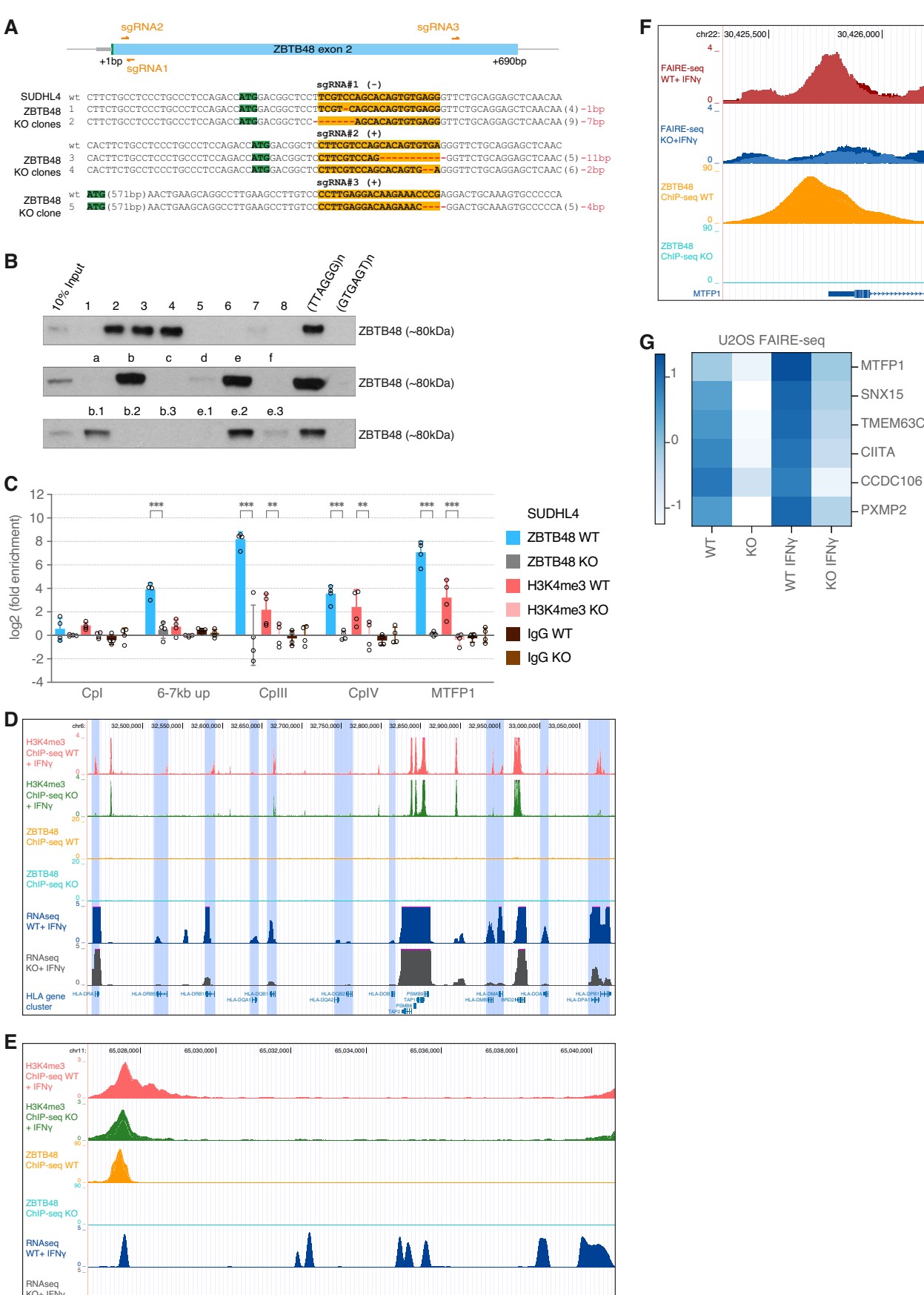

◀ **Figure EV5. ZBTB48 also binds to CIITA pIII and regulates H3K4me3 levels in the constitutively expressing DLBCL cell line SUDHL4.**

(A) Schematic of sgRNA design for the generation of ZBTB48 KO clones in SUDHL4. 5 different KO clones were identified and genotyped as shown. The start codon is shown in green and the sgRNA binding sites are highlighted in bold and yellow. (B) Western blot of DNA pull-down assay using SUDHL4 nuclear extracts demonstrates ZBTB48 binding at two sites within pIII. Telomeric (TTAGGG) and scramble control (GTGAGT) sequence are used as positive and negative controls, respectively. (C) ZBTB48 and H3K4me3 ChIP reactions from four independent SUDHL4 WT and ZBTB48 KO clones analyzed by qPCR for CIITA pI, pII (6–7 kb up), pIII, pIV. The MTFP1 promoter is used as a positive control. The data was normalized to IgG and gene desert region and fold change was calculated relative to the average of WT clones. Data represents mean ± SD. $p$ values were calculated by one-way ANOVA with Holm-Sidak correction for multiple comparisons ($n = 4$); *$p < 0.05$, **$p < 0.01$, ***$p < 0.001$. $p$ values for WT vs KO comparisons: for ZBTB48 enrichment: 6–7 kb up: $p = 4.397E-06$, CpIII: $p < 1E-15$, CpIV: $p = 2.986E-06$, MTFP1: $p < 1E-15$; for H3K4me3 enrichment: CpIII: $p = 0.009$, CpIV: $p = 0.003$, MTFP1: $p = 3.162E-06$. (D) ZBTB48 and H3K4me3 ChIP-seq tracks with concomitant RNA-seq tracks in U2OS WT and ZBTB48 KO clones +/− IFNγ treatment (250 U/ml IFNγ for 48 h) covering the MHC-II gene cluster. The location of individual HLA genes is highlighted in light blue across tracks. (E) ChIP-seq tracks depicting ZBTB48 at the SNX15 promoter in U2OS WT cells but not in the KO cells with concomitant loss of RNA expression as seen in the RNA-seq tracks. In contrast, the H3K4me3 peak is present at the SNX15 promoter in both U2OS WT and ZBTB48 KO clones. (F) FAIRE-seq tracks depicting open chromatin at the MTFP1 promoter in two independent IFNγ-induced U2OS WT cells but not in the KO cells. The ZBTB48 ChIP-seq tracks highlight the ZBTB48 binding region within the MTFP1 promoter. (G) Heatmap of FAIRE-seq profiles depicting the relative chromatin accessibility at ZBTB48 bound promoter in U2OS WT vs ZBTB48 KO cells +/− IFNγ treatment (250 U/ml IFNγ for 48 h). Intensity values are based on Z-scores of log2 rpkm values across samples. Source data are available online for this figure.

