## [Peer Review File · The EMBO Journal]

ZBTB48 is a priming factor regulating B-cell-specific CIITA expression

Grishma Rane, Vivian Kuan, Suman Wang, Michelle Mok, Vartika Khanchandani, Julia Hansen, Ieva Norvaisaite, Naasyidah Zulkaflee, Wai Yong, Arne Jahn, Vineeth Mukundan, Yunyu Shi, Motomi Osato, Fudong Li, and Dennis Kappei

Corresponding author(s): Dennis Kappei (dennis.kappei@nus.edu.sg)

Review Timeline:

Submission Date:	23rd Nov 23
Editorial Decision:	30th Mar 24
Revision Received:	12th Aug 24
Editorial Decision:	25th Sep 24
Revision Received:	26th Sep 24
Accepted:	16th Oct 24

Editor: Cornelius Schneider

Transaction Report:

Dear Dr Kappei,

Thank you again for submitting your manuscript to the EMBO Journal, for sharing your preliminary point-by-point reply and for our productive meeting. I have now contacted referees #1 and #3 again regarding your preliminary response to the comments of referee #2. Both referees find the proposed revisions adequate both regarding their own concerns and the points raised by referee #2. We would therefore like to invite you to revise your manuscript based on your preliminary point-by-point reply.

I should also add that it is The EMBO Journal policy to allow only a single major round of revision and that it is therefore important to resolve the main concerns at this stage.

We generally allow three months as standard revision time, which can be extended to 6 months in case of major revisions, such as the experiments required here. As a matter of policy, competing manuscripts published during this period will not negatively impact on our assessment of the conceptual advance presented by your study. However, we request that you contact the editor as soon as possible upon publication of any related work, to discuss how to proceed. Should you foresee a problem in meeting the deadline, please let us know in advance and we may be able to grant an extension.

Thank you for the opportunity to consider your work for publication. I look forward to your revision.

Yours sincerely,

Cornelius Schneider

Cornelius Schneider, PhD
Editor
The EMBO Journal
c.schneider@embojournal.org

We realize that it is difficult to revise to a specific deadline. In the interest of protecting the conceptual advance provided by the work, we recommend a revision within 3 months (28th Jun 2024). Please discuss the revision progress ahead of this time with the editor if you require more time to complete the revisions. Use the link below to submit your revision:

Referee #1:

In this manuscript, Rane et al showed that ZBTB48 is a priming factor for the promoter activation of the MHC class II coactivator CIITA in B cell specific manner. The study was motivated by ChIP-seq data in the previous study for ZBTB48 function on telomere, which was published by the same group in 2017. Although ARE (activation response element) sequences are known to be important upstream sequences for the expression of B cell-specific CIITA, the potential activators that bind to ARE sequences have been an open question. The authors demonstrated that ZBTB48 is an essential factor for the activation of pIII among CIITA promoters through binding to ARE sequences. This is an excellent study with well-designed experiments and analyses of data harboring both molecular mechanisms and in vivo phenotype of gene-deficient mice. Although the study is nearly complete, the manuscript would be improved by addressing the following points.

Major comments

1. While the author showed that ZBTB48 associated with CIITA pIII promoter, both pIII and pIV CIITA transcripts were reduced in ZBTB48-deficient cells in Figure 3 A. Figure 5B also showed altered histone methylation in the region of pIV in ZBTB48-deficient cells. It would be of interest if the role for ZBTB48 in the activation of pIII promoter is experimentally revealed, or at least discussed.
2. The role of ZBTB48 in CIITA promoter activation was found using epithelial cell lines, USO2 and Hela cells. Interestingly most prominent phenotype for MHC class II deficiency was observed in ZBTB48-deficient B cells, especially in pre- and immature B cells. The authors could strengthen their manuscript by discussing the mechanism how ZBTB48 functions in a B cell-specific manner.
3. Since the function of MHC class II is most significant in antigen presenting cells, MHC class II expression in major myeloid population such as macrophages, monocytes or dendritic cells should be evaluated in Figure 4F.

Minor points:

1. For Figure 4, the authors used the mouse model to test the function of ZBTB48 for MHC-II expression. The authors should comment on whether human and mouse ZBTB48 are conserved and can bind to the same sequence and can be used as a model.
2. Line 233 mentions Supplementary Fig. 3D and 3F, but the content probably refers to Supplementary Fig. 4. Similarly, line 242 also refers to Supplementary Fig. 4.
3. FACS methods or legend for Figure 4 should be improved for more detailed information. IgM D, E, F should be pre-B, immature B and mature B cells. What are IgD D, E, F population? IgD is, by definition, not expressed on immature B cells.

Referee #2:

This article presented by Grishma Rane et al., and entitled "ZBTB48 is a pioneer factor regulating B-cell-specific CIITA expression", describes the zinc finger protein that binds telomere regions ZBTB48 as a transcription regulator of CIITA, the major factor regulating MHCII expression. The manuscript is well written and the description of the results is accurate and rigorous. References cited seem to be appropriate and reflect well the relevant literature of the field that connects with this study. The authors claim that ZBTB48 controls the B-lymphocyte specific expression of CIITA by regulating the lymphoid-specific promoter (promoter III) of the CIITA gene. The study identifies a development transition in mouse primary B lymphocytes where ZBTB48 is required for MHCII protein expression. The authors also thoroughly characterize the mechanism of ZBTB48-dependent expression of CIITA in the osteosarcoma cell line U2OS. However, the study falls short in the characterization of the role of ZBTB in primary B lymphocytes, as well as in addressing its relevance to B cell biology. Most of the experiments in this work are done using U2OS cells, an immortalized human osteosarcoma cell line that has a high

level of chromosomal aberrations. If interested in the B lymphocyte -or simply lymphoid- specificity of CIITA expression, different human B cell lines could be used for all the experiments studying the mechanism by which ZBTB48 supports CIITA expression. The authors show data with SUDHL4 cells, an immortalized diffused large B cell lymphoma cell line expressing CIITA constitutively, but the study of the mechanism is for the most done using U2OS cells, which expresses CIITA upon IFN γ stimulation. For instance, in both cell lines ZBTB48 binds to promoter III and the non-hematopoietic-specific promoter IV of CIITA, albeit less to promoter IV; however, H3K4m3 deposition in both promoters is dependent on ZBTB48. It is interesting that the mechanism identified here for ZBTB48 binding in vitro to CIITA promoter III is different to the one it uses for binding to telomere regions. The authors also show a crystal structure that supports this novel mechanism. Considering that most of the work is done with IFN γ -treated U2OS cells, which are not of lymphoid origin; that expression of CIITA is induced by promoter IV in certain non-hematopoietic cells stimulated with IFN γ ; and considering also that the data shown in the study points to a certain degree of ZBTB48-dependent regulation of promoters III and IV in U2OS cells, one possibility is that the mechanism identified in U2OS is not specific for the B cell regulation of CIITA, and can facilitate expression of CIITA in non-hematopoietic cells (promoter IV). In this regard, it would be interesting to show whether human myeloid cell lines -which use promoter I- express CIITA in a manner independent of ZBTB48. Similarly, analysis of other non-hematopoietic cell lines that can express MHCII upon IFN γ stimulation, would address whether ZBTB48 can control CIITA and therefore MHCII. One important piece of information of this study is the analysis of MHCII expression in a mouse model deficient for ZBTB48 generated by CRISPR / Cas9 deletion. ZBTB48-deficient and control mice were analyzed by flow cytometry for the expression of MHCII in primary cells of the thymus, peripheral blood, spleen and bone marrow, including different developmental stages of B lymphocytes. This analysis of steady state populations is very intriguing as it reveals a contribution of ZBTB48 to CIITA expression in specific stages of B cell development, pre-B and immature B lymphocytes, and not so much in mature B cells. These results are relevant but are also quite preliminary in this manuscript. One can ask whether primary mature B lymphocytes would enhance their dependence on ZBTB48 for MHCII expression upon IFN γ stimulation. Also, the authors could isolate different populations of developing B cells and analyze their mRNA of CIITA. These experiments can help to make a better connection with the in vitro data they show using U2OS cells. Another important open question is the relevance of the role of ZBTB48 in regulating MHCII during specific stages of B cell development, and more so considering that the representation of the different B cell subsets during development is the same in ZBTB48-deficient and control mice. Also, based on the data presented, it is an overstatement that ZBTB48 acts as a pioneering transcription regulator. Pioneering transcription factors are frequently involved in promoting cell differentiation in development process. They work by displacing histones and facilitating recruitment of other transcription regulators. These functions are not studied for ZBTB48 in this work. In this regard, formaldehyde-assisted isolation of regulatory elements (FAIRE), a technical approach used in this work, informs about chromatin accessibility and not strictly histone displacement. The authors detect that, same as many non-pioneering transcription regulators, ZBTB facilitates chromatin accessibility in CIITA promoter III in U2OS cells. The authors state that ZBTB48 does not respond to IFN γ but the data presented do not rule out this possibility. The authors could test its level of expression by Western blot after IFN γ stimulation of U2OS cells. Even if no changes were detected after IFN γ stimulation, one cannot rule out that IFN γ could increase its activity by indirect means such as posttranslational modifications.

Minor points:

Some figures should be presented in a bigger size, as they really are too small to see. These are Figure 1A, Figure 3B, Figure 5C, Figure 6B, Supplementary Figures 6 D-F.

Regarding the graphs in supplementary figures 4 and 5, showing the percentage of different cellular populations in the ZBTB48 knockout, it is not what some of the percentages are referred to. For example, supplementary figure S5D should indicate that percentages of CD44 and CD25 are referred to DN cells. Same for S4D, S4E, S4F, S5A, and S5C.

Another minor point is when the authors write sentences including "promoter... expression". It seems more appropriate to refer to promoters as being activated rather than expressed.

Referee #3:

The manuscript by Rane et al. identifies the factor ZBTB48 as critical to MHC class II expression by regulating the transcriptional coactivator CIITA, the limiting factor regulating MHC-II expression. CIITA is expressed in a set of immune cells that typically present antigens and is inducible in nearly all other cell types following exposure to IFN-g. CIITA is regulated by 3 promoters that control tissue specific expression with pI functioning in myeloid cells, pIII in lymphoid cells, and pIV being used for most IFN γ responses in all other cell types. The authors discovered the role of ZBTB48 indirectly as they were studying ZBTB48 in other contexts and noted that ZBTB48 bound to the pIII region of CIITA and this binding was absent in the KO. In the current paper, the authors precisely determine the binding of ZBTB48 to the ARE sites in pIII using a series of binding assays and crystal structures of domains, they identify the Zn fingers of ZBTB48 that are important, and show that the protein is required for maximal induction of induction of CIITA by IFN γ and some MHC-II genes in U2OS cells. They also examine the role of ZBTB48 in vivo using their KO mouse where they show some changes in immune cell frequencies (bone marrow) and MHC-II expression. However, these changes in MHC-II are not across the board in all the cell subsets that they analyze; with some having massive changes in MHC-II expression in the KO. Lastly, they show that ZBTB48 induces changes in active histone marks and increases in accessibility using the FAIRE assay. From these combined data, they suggest that ZBTB48 is a pioneering factor that is required for all other factors to assemble.

Overall, the work here is extremely novel and exciting and should be published in EMBO J. For the most part is done extremely well. There are some concerns that the authors should address.

Major

1. Considering the potential for ZBTB48 regulating CIITA gene expression across multiple cell types, it is essential to know the levels of ZBTB48 expression in the cell types they are looking at. This could explain the discrepancy of MHC-II positive cells across development.
2. Fig sup 5 is somewhat surprising in that 1) the heterogeneity of MHC-II expression in HSPC subsets and 2) the expression of MHC-II on thymic T cells in the mouse. The SP 4s and 8s should not express MHC-II molecules. Is this background or true heterogeneity and why would these cells express these proteins?
3. Fig 3A and 3B don't quite agree in that the pan-CIITA is more than 50% of WT in 3A, yet not detectable in 3B?
4. Why did the authors not look at HLA-DRA, which is the highest of all MHC-II genes that are expressed? (Fig 3C). I could not find HLA-DRA (#20) on the volcano plot either? CIITA is clearly required for DRA expression.
5. In Fig 4C, why is MHC-II expression dependent on ZBTB48 in preB cells and not in mature? These inconsistencies are confusing and should be discussed or demonstrated experimentally.
6. Is ZBTB48 really a pioneer factor? This could be the case for the IFN γ induction system as described with the FAIRE experiments, but is this also true in B cells? ATAC-seq assays could provide an easy way to get at this.

Minor but still important

Nearly all of the figures/panels are unreadable in a printed version and require high blow ups when viewing on the computer. The readability reflects using gray (light reds / yellows/ orange) colors for text, thin lines that are not sharp, and way too small a font. This may require additional figures to be assembled given the space on a page, but would be worth it for the readability and examination of the data.

Reviewer: black

Authors: blue

Changes incorporated during revision: green

Referee #1 (Report for Author)

In this manuscript, Rane et al showed that ZBTB48 is a priming factor for the promoter activation of the MHC class II coactivator CIITA in B cell specific manner. The study was motivated by ChIP-seq data in the previous study for ZBTB48 function on telomere, which was published by the same group in 2017. Although ARE (activation response element) sequences are known to be important upstream sequences for the expression of B cell-specific CIITA, the potential activators that bind to ARE sequences have been an open question. The authors demonstrated that ZBTB48 is an essential factor for the activation of pIII among CIITA promoters through binding to ARE sequences. This is an excellent study with well-designed experiments and analyses of data harboring both molecular mechanisms and in vivo phenotype of gene-deficient mice. Although the study is nearly complete, the manuscript would be improved by addressing the following points.

We appreciate the referee's recognition of the quality of our study and the experimental plan, and thank the referee for the constructive comments below.

Major comments

1. While the author showed that ZBTB48 associated with CIITA pIII promoter, both pIII and pIV CIITA transcripts were reduced in ZBTB48-deficient cells in Figure 3 A. Figure 5B also showed altered histone methylation in the region of pIV in ZBTB48-deficient cells. It would be of interest if the role for ZBTB48 in the activation of pIII promoter is experimentally revealed, or at least discussed.

Concrete action: Textual improvements.

Indeed, the pIV transcript was 17-fold more highly expressed in WT compared to ZBTB48 KO clones (Fig. 3A) and the H3K4me3 histone mark spread across both pIII and pIV and is absent at both promoters in ZBTB48 KO clones (Fig. 5C). However, while our earlier ChIP-seq data had clearly identified a robust ZBTB48 peak at pIII, this was not the case at pIV, suggesting that ZBTB48 only directly binds to the pIII activating elements. This is in agreement with our ChIP-qPCR results in Fig. 5A in which we validated a >450 fold enrichment at pIII. While we did detect significant ZBTB48 enrichment at pIV, this was based on a comparably moderate 8-fold enrichment in parallel to a similar weak enrichment at a 6-7kb upstream region (corresponding to CIITA pII, which is not active and might rather correspond to an enhancer element; PMID: 17067677). These weak interactions observed by ChIP could be due to chromatin looping at the CIITA promoter region as described by Ni et al. in 2008 (PMID: 18500344). Therefore, one possible explanation is chromatin-chromatin interactions between these various elements that lead to the ZBTB48-dependent effect extending to pIV (in this cellular context).

To test this further, we have now measured pI, pIII and pIV CIITA promoter and MHC II transcript levels in five HeLa WT and ZBTB48 KO clones (new Fig. EV2B), similar to the previously existing data in U2OS cells (Fig. 3A,C). In HeLa cells both the pan-CIITA expression and CIITA pIV mRNA levels are unchanged between ZBTB48 WT and KO clones while CIITA pI was again not induced upon IFN γ treatment. In contrast, ZBTB48 KO clones only very moderately induced CIITA pIII levels upon IFN γ treatment with a 75-fold higher induction in HeLa WT clones. In combination with the ChIP-seq profile showing that ZBTB48 exclusively binds to CIITA pIII, we believe that our data support a model in which ZBTB48's primary and direct effect is on CIITA pIII while secondary, indirect affects (as seen by the reduction in CIITA pIV mRNA levels in U2OS cells) may be possible.

Fig. EV2B: Relative mRNA expression of total (pan-)CIITA and the three promoter-specific transcripts in each five independent HeLa WT and ZBTB48 KO clones treated with or without 250 U/ml IFN γ for 24 hours. Data represents mean \pm SEM. Data points with grey circles are imputed due to lack of detectable Ct values. The log₂ fold change is calculated relative to the average of five untreated WT clones. p-values were calculated by 2-way ANOVA (n = 5).

2. The role of ZBTB48 in CIITA promoter activation was found using epithelial cell lines, USO2 and Hela cells. Interestingly most prominent phenotype for MHC class II deficiency was observed in ZBTB48-deficient B-cells, especially in pre- and immature B-cells. The authors could strengthen their manuscript by discussing the mechanism how ZBTB48 functions in a B cell-specific manner.

Concrete action: Textual improvements.

We meant to say that ZBTB48 is regulating the B-cell specific promoter of CIITA rather than being unique to B-cells itself. Publicly available resources such as the Human Protein Atlas suggest that ZBTB48 is ubiquitously expressed with low tissue specificity. Therefore, our model entails that ZBTB48 acts as a priming factor that is constitutively bound to CIITA pIII if expressed. In non-APCs such as U2OS an additional stimulus in the form of IFN γ is required for other transcription factors to bind to the opened promoter and turn on CIITA expression while in B-cells we would expect such factors to be constitutively expressed and/or activated. We have adjusted the text to state this more explicitly:

"In such a model, ZBTB48 binding at nucleosome-compacted promoters induces a nucleosome-depleted region to subsequently grant access to other transcription factors and chromatin remodelling complexes. In the case of non-APCs this requires additional IFN γ stimulation and binding by IFN γ -dependent transcription factors while corresponding factors are likely constitutively expressed in B-cells. In both scenarios, this will subsequently solidify the open chromatin state and trigger transcription initiation by RNA polymerase II." (page 11 lines 333-339)

3. Since the function of MHC class II is most significant in antigen presenting cells, MHC class II expression in major myeloid population such as macrophages, monocytes or dendritic cells should be evaluated in Figure 4F.

Concrete action: Revised graphs in Fig. 4 and Fig. EV5 (previously Fig. EV4).

We had evaluated MHC II expression in myeloid cell populations in Fig. 4F,G and Fig. EV5D-G (e.g. Gr1-positive cells). However, we acknowledge that the original labelling only indicated

specific markers that were used to identify the myeloid cell populations, and is not entirely self-explanatory. We have now modified these figures to more clearly specify the corresponding cell types.

Minor points:

1. For Figure 4, the authors used the mouse model to test the function of ZBTB48 for MHC-II expression. The authors should comment on whether human and mouse ZBTB48 are conserved and can bind to the same sequence and can be used as a model.

Concrete action: **New Fig. EV5A & textual adjustment.**

Human and mouse ZBTB48 share 92% sequence identity, which we had previously analysed (Fig. 2B in PMID: 28176777). Importantly, most of the existing sequence variation is confined to the N-terminal region of ZBTB48 while ZnF10, ZnF11 and the C-terminal arm, that are required for direct binding to CIITA pIII (human residues 549-620 as in Fig. 2A), share 100% sequence identity between the human and mouse orthologs. In addition to documenting the sequence conservation of the ZBTB48-binding sites in both the human and mouse CIITA pIII (Fig. 4A), we have added this information in the text and we added the protein sequence alignment as a **new Fig. EV5A**.

Fig. EV5A: Protein sequence alignment using Clustal Omega of human and mouse ZBTB48 (top) with a zoom in to the DNA binding region including ZnF10, ZnF11 and the C-terminal arm that are required for binding to CIITA pIII (bottom).

“CIITA pIII is conserved in mice including the triple G motifs that are recognised by ZBTB48 through base-specific contacts within the b.1 and e.2 sequences (Fig. 4A). Likewise, the critical domains in ZBTB48 have 100% protein sequence identify between the human and mouse orthologs (Fig. EV5A), suggesting that ZBTB48 might confer similar regulation in mice.” (page 7 lines 218-222)

2. Line 233 mentions Supplementary Fig. 3D and 3F, but the content probably refers to Supplementary Fig. 4. Similarly, line 242 also refers to Supplementary Fig. 4.

Concrete action: Textual improvements.

We thank the referee for bringing this to our attention and we have corrected the text accordingly. The figure panels are now referred to as Fig. EV5E,F as well as Fig. EV5G,H.

3. FACS methods or legend for Figure 4 should be improved for more detailed information. IgM D, E, F should be pre-B, immature B and mature B cells. What are IgD D, E, F population? IgD is, by definition, not expressed on immature B cells.

Concrete action: Updated Fig. 4 & Fig. EV6.

The pre-B, immature and mature B-cells were gated twice independently, once using IgM and once using IgD to confirm our observations. Based on the referee’s comment we realised that the original representation might have been confusing. Therefore, we have moved the validation results from IgD gating for D (pre-B), E (immature) and F (mature) populations to Fig. EV5E,G. This figure originally had already contained corresponding representative FACS plots (Fig. EV5H).

Referee #2 (Report for Author)

This article presented by Grishma Rane et al., and entitled "ZBTB48 is a pioneer factor regulating B-cell-specific CIITA expression", describes the zinc finger protein that binds telomere regions ZBTB48 as a transcription regulator of CIITA, the major factor regulating MHCII expression. The manuscript is well written and the description of the results is accurate and rigorous. References cited seem to be appropriate and reflect well the relevant literature of the field that connects with this study.

We thank the referee for the positive assessment of our work and the helpful feedback below.

The authors claim that ZBTB48 controls the B-lymphocyte specific expression of CIITA by regulating the lymphoid-specific promoter (promoter III) of the CIITA gene. The study identifies a development transition in mouse primary B lymphocytes where ZBTB48 is required for MHCII protein expression. The authors also thoroughly characterize the mechanism of ZBTB48-dependent expression of CIITA in the osteosarcoma cell line U2OS. However, the study falls short in the characterization of the role of ZBTB in primary B lymphocytes, as well as in addressing its relevance to B cell biology.

Most of the experiments in this work are done using U2OS cells, an immortalized human osteosarcoma cell line that has a high level of chromosomal aberrations. If interested in the B lymphocyte -or simply lymphoid- specificity of CIITA expression, different human B cell lines could be used for all the experiments studying the mechanism by which ZBTB48 supports CIITA expression. The authors show data with SUDHL4 cells, an immortalized diffused large B cell lymphoma cell line expressing CIITA constitutively, but the study of the mechanism is for the most done using U2OS cells, which expresses CIITA upon IFN γ stimulation. For instance, in both cell lines ZBTB48 binds to promoter III and the non-hematopoietic-specific promoter IV of CIITA, albeit less to promoter IV; however, H3K4m3 deposition in both promoters is dependent on ZBTB48. It is interesting that the mechanism identified here for ZBTB48 binding in vitro to CIITA promoter III is different to the one it uses for binding to telomere regions. The authors also show a crystal structure that supports this novel mechanism.

We thank the referee for this clear summary of our work. We appreciate the concerns raised and address them below.

Considering that most of the work is done with IFN γ -treated U2OS cells, which are not of lymphoid origin; that expression of CIITA is induced by promoter IV in certain non-hematopoietic cells stimulated with IFN γ ; and considering also that the data shown in the study points to a certain degree of ZBTB48-dependent regulation of promoters III and IV in U2OS cells, one possibility is that the mechanism identified in U2OS is not specific for the B cell regulation of CIITA, and can facilitate expression of CIITA in non-hematopoietic cells (promoter IV).

We would like to refer to similar points raised by referee #1 above: While loss of ZBTB48 in U2OS cells also led to reduced induction of the CIITA pIV-specific transcript concomitant with loss of the H3K4me3 signal spanning both promoters, this is likely an indirect effect given that ZBTB48 only robustly binds to the two ARE elements in CIITA pIII. In addition, in our ZBTB48 KO mouse model only B-cells were affected in their MHC II expression levels, again suggesting that ZBTB48 primarily affects CIITA pIII (and/or other factors more prominently regulate CIITA pI and pIV in other lineages).

In this regard, it would be interesting to show whether human myeloid cell lines -which use promoter I- express CIITA in a manner independent of ZBTB48. Similarly, analysis of other

non-hematopoietic cell lines that can express MHCII upon IFN γ stimulation, would address whether ZBTB48 can control CIITA and therefore MHCII.

Concrete action: RT-qPCR to detect CIITA and MHC II levels in HeLa cells as a second non-APC cell line

As mentioned above, we did not detect any significant changes in the percentage of MHC II positive cells in the myeloid cell population from spleen cells (Fig. 4F), bone marrow (Fig. 4G) and blood (Fig. EV4) in Zbtb48 KO mice in comparison to WT mice. To complement these data, we have profiled 10 additional myeloid cell lines. However, across all of these cell lines CIITA pI was either undetectable or faint and either CIITA pIII or pIV were the predominant isoforms. Therefore, we are not able to directly test that CIITA pI is unaffected in the absence of ZBTB48.

Figure to the reviewers #1: Relative mRNA expression of total (pan-)CIITA and the three promoter-specific transcripts in each ten myeloid cell lines with or without 250 U/ml IFN γ for 24 hours. Fold change is calculated relative to an imputed Ct value of 35 (as a conservative estimate in reactions without detectable expression).

However, the referee raises an important point about using additional non-APC cell lines. We have now measured pI, pIII and pIV CIITA promoter and MHC II transcript levels in five HeLa WT and ZBTB48 KO clones (new Fig. EV2B), similar to the previously existing data in U2OS cells (Fig. 3A,C). In HeLa cells both the pan-CIITA expression and CIITA pIV mRNA levels are unchanged between ZBTB48 WT and KO clones while CIITA pI was again not induced upon IFN γ treatment. In contrast, ZBTB48 KO clones only very moderately induced CIITA pIII levels upon IFN γ treatment with a 75-fold higher induction in HeLa WT clones. In combination with the ChIP-seq profile showing that ZBTB48 exclusively binds to CIITA pIII, we believe that our data support a model in which ZBTB48's primary and direct effect is on CIITA pIII while secondary, indirect affects (as seen by the reduction in CIITA pIV mRNA levels in U2OS cells) may be possible.

Fig. EV2B: Relative mRNA expression of total (*pan-*)CIITA and the three promoter-specific transcripts in each five independent HeLa WT and ZBTB48 KO clones treated with or without 250 U/ml IFN γ for 24 hours. Data represents mean \pm SEM. Data points with grey circles are imputed due to lack of detectable Ct values. The log₂ fold change is calculated relative to the average of five untreated WT clones. *p*-values were calculated by 2-way ANOVA (*n* = 5).

One important piece of information of this study is the analysis of MHCII expression in a mouse model deficient for ZBTB48 generated by CRISPR / Cas9 deletion. ZBTB48-deficient and control mice were analyzed by flow cytometry for the expression of MHCII in primary cells of the thymus, peripheral blood, spleen and bone marrow, including different developmental stages of B lymphocytes. This analysis of steady state populations is very intriguing as it reveals a contribution of ZBTB48 to CIITA expression in specific stages of B cell development, pre-B and immature B lymphocytes, and not so much in mature B cells.

We again thank the referee for this clear summary of the *in vivo* parts of our study.

These results are relevant but are also quite preliminary in this manuscript. One can ask whether primary mature B lymphocytes would enhance their dependence on ZBTB48 for MHCII expression upon IFN γ stimulation. Also, the authors could isolate different populations of developing B cells and analyze their mRNA of CIITA. These experiments can help to make a better connection with the *in vitro* data they show using U2OS cells. Another important open question is the relevance of the role of ZBTB48 in regulating MHCII during specific stages of B cell development, and more so considering that the representation of the different B cell subsets during development is the same in ZBTB48-deficient and control mice.

Concrete action: qPCR in developmental pre-B, immature and mature B cells to detect CIITA pI, pIII and pIV transcripts.

We agree with the referee that our identification of ZBTB48 as a critical CIITA pIII priming factor and the *in vivo* effects on primary B cells in our newly established Zbtb48 KO mouse model raise many new, interesting research questions. Based on a similar point by referee #3 below, we have now profiled both Zbtb48 and CIITA mRNA expression levels across developmental B cell stages in 12-16 weeks old wild-type mice to better understand the stage-specific regulation of CIITA expression in the B-cell lineage. High pan-CIITA mRNA levels are seen across the pre-B, immature and mature developmental B-cell stages and are directly mirrored by CIITA pIII mRNA levels with a moderate 4-fold increase in the mature populations. Concomitantly, ZBTB48 mRNA levels remain constant across B-cell development. While as expected CIITA pI levels are close to the qPCR detection limit, there is a gradual increase in

CIITA pIV (albeit at low to moderate levels compared to CIITA pIII). These data have now been integrated as a **new Fig. EV6I**.

Fig. EV6I: Relative mRNA expression of total (pan-)CIITA, the three promoter-specific transcripts and ZBTB48 mRNA in D (pre-B), E (immature) and F (mature) subpopulations from WT mice ($n = 3$). Data represents mean \pm SEM. The \log_2 fold change is calculated relative to CIITA pI in the mature subpopulation. p -values were calculated by 2-way ANOVA.

Beyond the very valid points raised by the referee, there are several alternative possibilities. For instance, loss of Zbtb48 could trigger promoter switching as a compensatory mechanism. Alternatively, given that even in pre-B populations a small fraction (<5% of cells) remain MHC II-positive, quality control mechanisms might lead to selection of these cells, explaining the smaller effect in mature populations. In such a scenario, one could envision that phenotypes aggravate with age. We hope that the referee agrees that the entirety of these experiments warrant an independent follow-up study, not least given the substantial effort and time requirements for the associated mouse work.

We discuss this in our manuscript accordingly (**page 12, lines 357-367**):

“These data suggest that there is heterogeneity in the B-cell population either in their CIITA promoter choice and/or in terms of expression of potentially redundant regulators. Even in systemic CIITA knock-out mice, 1-3% MHC II-positive splenic B-cells were detected³⁵, implying that the small portion of ZBTB48-independent, MHC II-positive pre-B-cells might be selected for and populate the mature B-cell stages. While at present we cannot explain why only specifically female ZBTB48^{-/-} mice present with a mild splenomegaly, quality control mechanisms triaging out MHC II-negative precursors, might contribute to the observed phenotype. Such a scenario should ultimately lead to stem cell exhaustion in the B-cell lineage with increasingly smaller numbers of MHC-positive cells in mature B cell populations and/or a reduction in the corresponding cell numbers.”

Also, based on the data presented, it is an overstatement that ZBTB48 acts as a pioneering transcription regulator. Pioneering transcription factors are frequently involved in promoting cell differentiation in development process. They work by displacing histones and facilitating recruitment of other transcription regulators. These functions are not studied for ZBTB48 in this work. In this regard, formaldehyde-assisted isolation of regulatory elements (FAIRE), a technical approach used in this work, informs about chromatin accessibility and not strictly histone displacement. The authors detect that, same as many non-pioneering transcription regulators, ZBTB facilitates chromatin accessibility in CIITA promoter III in U2OS cells.

Concrete action: Change wording to “priming factor”

The reviewer raises important points and we agree that while our data are consistent with a putative pioneering activity, the terminology comes with the expectation of developmental regulation (which is unlikely for ZBTB48 given a relatively ubiquitous expression profile across tissues) and a more detailed understanding of the precise molecular steps (e.g. direct binding to nucleosomal DNA, histone displacement, facilitated recruitment of other transcription regulators/chromatin remodellers) will be needed in future work, which we now highlight in the discussion (**page 13, lines 381-386**):

“Upon opening of the locus, CIITA pIII would subsequently become accessible for other transcription factors that in turn are required to trigger gene expression (**Fig. 6C**). It will be important in future work to establish how ZBTB48 promotes histone displacement and how it may work in concert with other factors. Meanwhile, ZBTB48 could be equated to an on-off light switch while additional transcription factors function as dimmer switches.”

In addition, we have adjusted the wording throughout the manuscript to priming factor (see wording by referee #1).

The authors state that ZBTB48 does not respond to IFN γ but the data presented do not rule out this possibility. The authors could test its level of expression by Western blot after IFN γ stimulation of U2OS cells. Even if no changes were detected after IFN γ stimulation, one cannot rule out that IFN γ could increase its activity by indirect means such as posttranslational modifications.

Concrete action: Textual adjustment.

We had previously measured ZBTB48 protein levels by Western blot upon IFN γ treatment and did not detect differences with untreated cells (**Fig. 3E**). In addition, we also did not detect significant changes in the enrichment of ZBTB48 at pIII upon IFN γ treatment by ChIP-qPCR (**Fig. 5A**). Nevertheless, we agree that there remains the possibility of IFN γ -dependent posttranslational modifications that could alter the function of ZBTB48. We have adjusted the text in the results section accordingly to reflect this possibility (**page 7, lines 195-199**):

“These results suggest that the impact of ZBTB48 in response to IFN γ stimulation is exclusive to CIITA expression and that ZBTB48 itself likely does not respond to IFN γ , even though at present we cannot exclude the possibility of IFN γ -induced post-translational modifications contributing to the ZBTB48-dependent CIITA activation.”

Minor points:

Concrete action: Adjustments to figures and text.

Some figures should be presented in a bigger size, as they really are too small to see. These are Figure 1A, Figure 3B, Figure 5C, Figure 6B, Supplementary Figures 6 D-F.

We apologise for this. We had originally exported the genome browser snapshots as non-editable jpg files. We have now replaced all tracks with better quality images and improved the labelling.

Regarding the graphs in supplementary figures 4 and 5, showing the percentage of different cellular populations in the ZBTB48 knockout, it is not what some of the percentages are referred to. For example, supplementary figure S5D should indicate that percentages of CD44 and CD25 are referred to DN cells. Same for S4D, S4E, S4F, S5A, and S5C.

The graph currently indicates the markers that were used to identify the myeloid cell populations, which we agree is not self-explanatory and might be confusing. We have now modified the figures to also indicate the cell types in addition to the individual markers.

Another minor point is when the authors write sentences including "promoter... expression". It seems more appropriate to refer to promoters as being activated rather than expressed.

We do note that we write e.g. "CIITA pIII expression". However, we are here referring to the promoter III-specific transcript of CIITA in line with how these isoforms have been referred to in the literature.

Referee #3 (Report for Author)

The manuscript by Rane et al. identifies the factor ZBTB48 as critical to MHC class II expression by regulating the transcriptional coactivator CIITA, the limiting factor regulating MHC-II expression. CIITA is expressed in a set of immune cells that typically present antigens and is inducible in nearly all other cell types following exposure to IFN-g. CIITA is regulated by 3 promoters that control tissue specific expression with pI functioning in myeloid cells, pIII in lymphoid cells, and pIV being used for most IFN-g responses in all other cell types. The authors discovered the role of ZBTB48 indirectly as they were studying ZBTB48 in other contexts and noted that ZBTB48 bound to the pIII region of CIITA and this binding was absent in the KO. In the current paper, the authors precisely determine the binding of ZBTB48 to the ARE sites in pIII using a series of binding assays and crystal structures of domains, they identify the Zn fingers of ZBTB48 that are important, and show that the protein is required for maximal induction of induction of CIITA by IFN-g and some MHC-II genes in U2OS cells. They also examine the role of ZBTB48 in vivo using their KO mouse where they show some changes in immune cell frequencies (bone marrow) and MHC-II expression. However, these changes in MHC-II are not across the board in all the cell subsets that they analyze; with some having massive changes in MHC-II expression in the KO. Lastly, they show that ZBTB48 induces changes in active histone marks and increases in accessibility using the FAIRE assay. From these combined data, they suggest that ZBTB48 is a pioneering factor that is required for all other factors to assemble.

Overall, the work here is extremely novel and exciting and should be published in EMBO J. For the most part is done extremely well. There are some concerns that the authors should address.

We thank the referee for endorsing our manuscript and the insightful comments.

Major

1. Considering the potential for ZBTB48 regulating CIITA gene expression across multiple cell types, it is essential to know the levels of ZBTB48 expression in the cell types they are looking at. This could explain the discrepancy of MHC-II positive cells across development.

Concrete action: qPCR in developmental pre-B, immature and mature B-cells to detect CIITA pI, pIII and pIV transcripts.

In line with a similar point raised by referee #2, we have now performed qPCR analysis of ZBTB48 (and transcript-specific CIITA) expression levels in the pre-B, immature and mature B-cell population in WT mice. High Pan-CIITA mRNA levels were seen across the developmental B-cell stages and are directly mirrored by CIITA pIII mRNA levels with a moderate 4-fold increase in the mature populations. Concomitantly, ZBTB48 mRNA levels remain constant across B-cell development. While as expected CIITA pI levels are close to the qPCR detection limit, there is a gradual increase in CIITA pIV (albeit at low to moderate levels compared to CIITA pIII). These data have now been integrated as a **new Fig. EV6I**.

Fig. EV6I: Relative mRNA expression of total (pan-)CIITA, the three promoter-specific transcripts and ZBTB48 mRNA in D (pre-B), E (immature) and F (mature) subpopulations from WT mice ($n = 3$). Data represents mean \pm SEM. The \log_2 fold change is calculated relative to CIITA pI in the mature subpopulation. p -values were calculated by 2-way ANOVA.

2. Fig sup 5 is somewhat surprising in that 1) the heterogeneity of MHC-II expression in HSPC subsets and 2) the expression of MHC-II on thymic T cells in the mouse. The SP 4s and 8s should not express MHC-II molecules. Is this background or true heterogeneity and why would these cells express these proteins?

Concrete action: Figure to the reviewer.

The FACS data are based on a widely used pan-MHC II antibody (I-A/I-E; clone M5) to determine MHC II-positive percentages across different cell populations above background signal using compensation beads to set gating parameters. As shown by representative FACS plots for the CD4 and CD8 single-positive cells below, the signal is robust and looks genuine. To exclude additional technical caveats, we would require a constitutive CIITA knock-out mouse to establish signal specificity across all cell types. Furthermore, MHC II expression on CD4 and CD8 developing T cells in thymus as well as on HSPC subsets has previously been reported (PMID: 16226503, PMID: 35523139). As for data heterogeneity, this largely correlates with the absolute abundance of cell types, i.e. more heterogeneity is seen in smaller populations. While we acknowledge these concerns, we would like to stress that the focus of our analyses here is primarily whether any cell types other than B cells present a ZBTB48-dependent difference in MHC II expression.

Figure to the reviewers #2: Representative flow cytometry analysis of MHC II positive cells in CD4+ and CD8+ SP cells from WT, Het and KO littermates.

3. Fig 3A and 3B don't quite agree in that the pan-CIITA is more than 50% of WT in 3A, yet not detectable in 3B?

Concrete action: Figure to the reviewer.

This might be a minor misunderstanding. The pan-CIITA levels are 34-fold higher in the U2OS WT cells upon IFN γ treatment compared to ZBTB48 KO clones. Please note that the data in **Fig. 3A** are shown in log₂ scale while the RNA-seq tracks in **Fig. 3B** have a linear y-axis. We chose log₂ scale for our qPCR data in parts to be able to visualize more lowly expressed transcripts. For reference, we replotted **Fig. 3A** in linear scale below, which suppresses more nuanced aspects, e.g. changes in CIITA pIV are no longer visible.

Figure to the reviewers #3: Data in Fig. 3A replotted in linear scale.

4. Why did the authors not look at HLA-DRA, which is the highest of all MHC-II genes that are expressed? (Fig 3C). I could not find HLA-DRA (#20) on the volcano plot either? CIITA is clearly required for DRA expression.

Concrete action: Edit **Fig. 3D** and **new Fig. EV3B**.

We originally performed qPCR for a representative set of HLA genes for **Fig. 3C** while the RNA-seq data in **Fig. 3D** contains all MHC II related transcripts. Thanks to this point, we realised that the HLA-DRA data point in **Fig. 3D** (#20) overlapped with another transcript (HLA-DRB1, #19) and we have adjusted the labelling of both transcripts on the volcano plot. To further illustrate this, we have generated a heatmap focused on the MHC II related transcripts only, which we have included as a **new Fig. EV3B**.

Fig. 3EV3B: Heatmap of differentially expressed MHC II family genes detected in RNA-seq data comparing 5 U2OS WT and ZBTB48 KO clones +/- IFN γ .

In addition, we would like to highlight that HLA-DRA is the most differentially expressed protein in the volcano plot in **Fig. 3F**, where we compared proteomes of IFN γ treated WT and ZBTB48 KO clones using label-free quantitative mass spectrometry. Similar to the RNA-seq data, we have added a new heatmap focused on the detected/quantified MHC II related proteins from our global proteome analysis as **Fig. EV4E**.

Fig. EV4E: Heatmap of differentially expressed MHC II family genes detected in the proteome data comparing 5 U2OS WT and ZBTB48 KO clones +/- IFN γ .

5. In Fig 4C, why is MHC-II expression dependent on ZBTB48 in preB cells and not in mature? These inconsistencies are confusing and should be discussed or demonstrated experimentally.

The near absence of MHC II positive pre-B-cells in our Zbtb48 KO mice tapered off as the cells developed into mature B-cells. We agree with the referee that it is intriguing that the loss of ZBTB48-dependent MHC II-positive cells is restricted to certain stages of B-cell development. In addition to related considerations by referee #2, we can think of three possible explanations: (1) Redundancy with other CIITA pIII regulators in more mature B-cell populations, (2) Promoter switching as a compensatory mechanism or (3) given that even in pre-B populations a small fraction (<5% of cells) remain MHC II-positive, quality control mechanisms might lead to selection of these cells which then populate the mature B-cell stages in the ZBTB48 KO mice. This might eventually lead to exhaustion of the MHC II positive B cell pool and/or stem cell exhaustion as the mice age.

We discuss this in our manuscript accordingly (**page 12, lines 357-367**):

“These data suggest that there is heterogeneity in the B-cell population either in their CIITA promoter choice and/or in terms of expression of potentially redundant regulators. Even in systemic CIITA knock-out mice, 1-3% MHC II-positive splenic B-cells were detected³⁵, implying that the small portion of ZBTB48-independent, MHC II-positive pre-B-cells might be selected for and populate the mature B-cell stages. While at present we cannot explain why only specifically female ZBTB48^{-/-} mice present with a mild splenomegaly, quality control mechanisms triaging out MHC II-negative precursors, might contribute to the observed phenotype. Such a scenario should ultimately lead to stem cell exhaustion in the B-cell lineage with increasingly smaller numbers of MHC-positive cells in mature B cell populations and/or a reduction in the corresponding cell numbers.”

6. Is ZBTB48 really a pioneer factor? This could be the case for the IFN γ induction system as described with the FAIRE experiments, but is this also true in B cells? ATAC-seq assays could provide an easy way to get at this.

Concrete action: Change wording to “priming factor”

We agree that ATAC-seq in primary B cells would be the ideal experiment to strengthen our model further. Despite the limiting number of pre-B cells retrieved by FACS sorting (~100k), we have indeed attempted this experiment given that pre-B cells show the strongest reduction in MHC II-positive cells in ZBTB48^{-/-} mice. However, even across several WT animals the ATAC-seq profiles were highly heterogenous. Given that animal experiments take several months to be repeated, we here propose to change the wording to “priming factor” (see wording by referee #1). This allows us to still clearly describe the main finding of our current

work while remaining cautious towards expectations that come with the “pioneer” terminology (e.g. see a similar comment by referee #2) and to more systematically study various aspects related to pioneering activity (e.g. direct binding to nucleosomal DNA, histone displacement, facilitated recruitment of other transcription regulators/chromatin remodellers) in follow-up work. We now highlight the n in future work (**page 13, lines 381-386**):

“Upon opening of the locus, CIITA pIII would subsequently become accessible for other transcription factors that in turn are required to trigger gene expression (**Fig. 6C**). It will be important in future work to establish how ZBTB48 promotes histone displacement and how it may work in concert with other factors. Meanwhile, ZBTB48 could be equated to an on-off light switch while additional transcription factors function as dimmer switches.”

Minor but still important

Nearly all of the figures/panels are unreadable in a printed version and require high blow ups when viewing on the computer. The readability reflects using gray (light reds / yellows/ orange) colors for text, thin lines that are not sharp, and way too small a font. This may require additional figures to be assembled given the space on a page, but would be worth it for the readability and examination of the data.

Concrete action: Figure and textual adjustment.

Based on a similar comment by referee #2, we assume that this comment is primarily referring to the genome browser snapshots in Fig. 1A, 3B, 5C, 6B and EV6C-E. We had originally exported the genome browser snapshots as non-editable jpg files. We have now replaced all tracks with better quality images and improved the labelling.

Dear Dr. Kappei,

Thank you for submitting your manuscript for consideration by the EMBO Journal. It has now been seen again by all three referees whose comments are enclosed below. As you will see, referees #1 and #3 find all of their concerns to be addressed and recommend publication of the manuscript. Referee #2 thinks that most concerns have been addressed but also remarks that there are two important points which still need revisions. We find both of these concerns reasonable and we would therefore like to invite you to submit a revised version of the manuscript, addressing these remaining concerns by referee #2.

Please do not hesitate to contact me if you have any additional questions.

We generally allow three months as standard revision time. As a matter of policy, competing manuscripts published during this period will not negatively impact on our assessment of the conceptual advance presented by your study. However, we request that you contact the editor as soon as possible upon publication of any related work, to discuss how to proceed.

Thank you for the opportunity to consider your work for publication. I look forward to your revision.

Yours sincerely,

Cornelius Schneider, PhD
Editor
The EMBO Journal
c.schneider@embojournal.org

We realize that it is difficult to revise to a specific deadline. In the interest of protecting the conceptual advance provided by the work, we recommend a revision within 3 months (24th Dec 2024). Please discuss the revision progress ahead of this time with

the editor if you require more time to complete the revisions. Use the link below to submit your revision:

Referee #1:

This reviewer has no further criticism on this revised manuscript.

Referee #2:

While I still consider that two points need further work to consolidate important messages of this manuscript (Rane et al. "ZBTB48 is a priming factor regulating B-cell-specific CIITA expression"), the authors have improved their manuscript and now provide more solid information in support of the role of ZBTB48 in the expression of CIITA in B lymphocytes.

Major points

1. One point to consolidate is connected with the experiments done with additional non-APC cell lines. As these experiments show that ZBTB48 KO clones only moderately induced CIITA pIII levels upon IFN gamma treatment with a 75-fold higher induction in WT clones, these findings show that ZBTB48 also regulates IFN gamma-induced expression of CIITA in cells different from B lymphocytes via pIII. This information, and also the information regarding the ZBTB48-independent regulation of pIV, should be incorporated into the discussion. Seeing the findings in the article, the message that ZBTB48 regulates CIITA expression in B cells is perhaps too narrow, so the discussion should include more context about other scenarios considering different cell types, different promoters and additional IFN gamma stimulation.

2. The other point is the analysis of the expression (mRNA) ZBTB48-dependent of different CTIIA forms and MHCII genes in primary B lymphocytes, both during B cell development and also in mature B lymphocytes untreated or IFN gamma-treated. Being ZBTB48 a transcription factor, and despite having analyzed MHCII protein by flow cytometry, it is very important to show the mRNA levels of MHCII and particularly the transcripts of the different CIITA promoters in WT and KO mice. IFN gamma-treated B cells could be cultured in vitro with or without LPS (as LPS expands B cells). This point is relevant to truly link findings in the article done in vitro with cell lines, with the true primary cells.

Minor points

Page 6, line 191: the figures cited here are probably Fig. EV2C,D and not Fig. 2C,D

The EV4 figure includes panel E, however this figure is not cited in the manuscript.

Page 8, lines 253 and 254: the parenthesis cites Supplementary Figure 6A and 6B, but it is most likely EV6A,B

Referee #3:

The authors have addressed my concerns.

Reviewer: black

Authors: blue

Changes incorporated during revision: green

Referee #1:

This reviewer has no further criticism on this revised manuscript.

Referee #3:

The authors have addressed my concerns.

We thank both referees for the positive assessment of our revision and supporting the publication of our work.

Referee #2:

While I still consider that two points need further work to consolidate important messages of this manuscript (Rane et al. "ZBTB48 is a priming factor regulating B-cell-specific CIITA expression"), the authors have improved their manuscript and now provide more solid information in support of the role of ZBTB48 in the expression of CIITA in B lymphocytes.

We thank the referee for the positive assessment of our revision.

Major points

1. One point to consolidate is connected with the experiments done with additional non-APC cell lines. As these experiments show that ZBTB48 KO clones only moderately induced CIITA pIII levels upon IFN gamma treatment with a 75-fold higher induction in WT clones, these findings show that ZBTB48 also regulates IFN gamma-induced expression of CIITA in cells different from B lymphocytes via pIII. This information, and also the information regarding the ZBTB48-independent regulation of pIV, should be incorporated into the discussion. Seeing the findings in the article, the message that ZBTB48 regulates CIITA expression in B cells is perhaps too narrow, so the discussion should include more context about other scenarios considering different cell types, different promoters and additional IFN gamma stimulation.

We thank the reviewer for this feedback and we want to clarify that since the original manuscript submission our message has been that ZBTB48 regulates CIITA pIII expression both for constitutive expression in B cells as well as in non-APCs upon IFN γ induction. To clarify this further we have expanded both the beginning and end of our introduction:

"The precisely fine-tuned and strictly regulated cell type specific expression of MHC II is primarily regulated at the transcriptional level by the differential usage of the three independent promoters of CIITA. In addition to being IFN γ -inducible, pIII predominantly drives the constitutive expression of CIITA in B cells. Here, we describe ZBTB48 as a key regulator of both IFN γ -inducible and constitutive *in vivo* expression of CIITA pIII." (page 13, lines 364-369).

"Upon opening of the locus, CIITA pIII would subsequently become accessible for other transcription factors that in turn are required to trigger gene expression (Fig. 6C). In the case of B cells these factors are likely constitutively expressed while in non-APCs their activity requires IFN γ stimulation in addition to the pre-existing ZBTB48-dependent CIITA pIII priming. It will be important in future work to establish how ZBTB48 promotes histone displacement, how it may work in concert with other factors and how CIITA pIII regulation may spillover to

other CIITA promoters as seen with CIITA pIV in U2OS cells (**Fig. 3A**). Meanwhile, ZBTB48 could be equated to an on-off light switch while additional transcription factors function as dimmer switches. Hence, while ZBTB48 determines whether gene activation is possible at all, the scale of transcriptional output is regulated downstream of the priming activity by other factors whose own activity limits CIITA pIII expression to B-cells and non-APCs upon IFN γ induction." (**page 14, lines 409-421**).

2. The other point is the analysis of the expression (mRNA) ZBTB48-dependent of different CTIIA forms and MHCII genes in primary B lymphocytes, both during B cell development and also in mature B lymphocytes untreated or IFN gamma-treated. Being ZBTB48 a transcription factor, and despite having analyzed MHCII protein by flow cytometry, it is very important to show the mRNA levels of MHCII and particularly the transcripts of the different CIITA promoters in WT and KO mice. IFN gamma-treated B cells could be cultured in vitro with or without LPS (as LPS expands B cells). This point is relevant to truly link findings in the article done in vitro with cell lines, with the true primary cells.

We agree with the reviewer that an experimentally testable expectation of our model is that in ZBTB48 KO pre-B cells both CIITA pIII as well as MHC II mRNA levels are depleted. In our view, we have provided compelling evidence for this model:

1. We have established the exact direct binding sites for ZBTB48 in CIITA pIII (**Fig. 1 and 2**). Both the binding sites (**Fig. 4A**) and the critical zinc fingers 10 & 11 in ZBTB48 (**Fig. EV3A**) are conserved between mice and human.
2. We have extensively documented the direct transcriptional impact on the CIITA pIII transcript as well as downstream MHC II mRNA expression levels in human non-APCs (**Fig. 3, EV1**).
3. We have profiled MHC II expression by FACS across different blood lineages in spleen, bone marrow and specifically examined developmental B cells and we demonstrate that only B-cells, with the most pronounced phenotype in pre-B cells, show a loss of MHC II expression (**Fig. 4, EV4**).
4. Based on the reviewer's suggestion, we had further validated that CIITA pIII is the primary transcript in developmental B cells (**Fig. EV4I**) in line with the well-established B-cell-specific expression pattern that had been established in the literature (PMID: 9184229, 12133965 & 12218128).

While the proposed experiments would add one more line of evidence to what we believe is an already solidly established finding, the delay would be substantial due to the need to cross animals and mature them to the matching time window, to administratively amend our IACUC protocol to include culturing primary cells and to newly establish primary B-cell culture in our laboratory. We are therefore of the opinion that these experiments may be better incorporated in a follow-up study, which will have to carefully evaluate how overall MHC II expression (partially) recovers upon B-cell maturation, e.g. through promoter switching, heterogeneity in the B-cell compartment and/or quality control mechanisms selecting MHC II-positive pre-B-cells for maturation (see our discussion **page 13, lines 383-401**).

Minor points

Page 6, line 191: the figures cited here are probably Fig. EV2C,D and not Fig. 2C,D

The EV4 figure includes panel E, however this figure is not cited in the manuscript.

Page 8, lines 253 and 254: the parenthesis cites Supplementary Figure 6A and 6B, but it is most likely EV6A,B

Thank you for spotting these inaccuracies, which we have now corrected.

Dear Dr. Kappei,

I am pleased to inform you that your manuscript has been accepted for publication in the EMBO Journal.

Yours sincerely,

Cornelius Schneider

Cornelius Schneider, PhD
Editor
The EMBO Journal
c.schneider@embojournal.org
